# PEMFC Current Control Using a Novel Compound Controller Enhanced by the Black Widow Algorithm: A Comprehensive Simulation Study

Mohammed Yousri Silaa [1,2,*], Oscar Barambones [2,*], José Antonio Cortajarena [3], Patxi Alkorta [3] and Aissa Bencherif [2]

1   Telecommunications Signals and Systems Laboratory (TSS), Amar Telidji University of Laghouat, BP 37G, Laghouat 03000, Algeria
2   Engineering School of Vitoria, University of the Basque Country UPV/EHU, Nieves Cano 12, 1006 Vitoria, Spain; a.bencherif@lagh-univ.dz
3   Engineering School of Gipuzkoa, University of the Basque Country, UPV/EHU, Avda Otaola N29, 20600 Eibar, Spain; josean.cortajarena@ehu.eus (J.A.C.); patxi.alkorta@ehu.eus (P.A.)
*   Correspondence: moh.silaa@lagh-univ.dz or silaa.mohammed.yousri@gmail.com (M.Y.S.); oscar.barambones@ehu.eus (O.B.)

**Abstract:** Proton exchange membrane fuel cells (PEMFCs) play a crucial role in clean energy systems. Effective control of these systems is essential to optimize their performance. However, conventional control methods exhibit limitations in handling disturbances and ensuring robust control. To address these challenges, this paper presents a novel PI sliding mode controller-based super-twisting algorithm (PISMCSTA). The proposed controller is applied to drive the DC/DC boost converter in order to improve the PEMFC output power quality. In addition, the black widow optimization algorithm (BWOA) has been chosen to enhance and tune the PISMCSTA parameters according to the disturbance changes. The performance of the PISMCSTA is compared with the conventional STA controller. Comparative results are obtained from numerical simulations and these results show that the developed proposed PISMCSTA gives better results when compared to the conventional STA. A reduction of up to 8.7% in the response time could be achieved and up to 66% of the chattering effect could be eliminated by using the proposed controller. Finally, according to these results, the proposed approach can offer an improvement in energy consumption.

**Keywords:** PEMFC; PI controller; DC/DC boost converter; SMC; STA; BWOA

## 1. Introduction

### 1.1. Motivations

In recent years, the scientific community has become increasingly alarmed by a series of indicators pointing to the severity of climate change. Factors such as accelerating ocean warming, heatwaves in unexpected locations like Siberia, and uncontrollable wildfires in the world's primary forests all underscore the urgency of addressing climate change [1]. The emission of greenhouse gases, particularly carbon dioxide ($CO_2$), is frequently cited as a primary contributing factor. The impact of climate change poses a significant threat to humanity, largely attributed to the continued use of fossil fuels and coal [2]. To ensure the survival of our species, we must replace traditional fuels with renewable ones such as wind [3], solar power [4], bio-oil [5], and hydrogen power sources [6]. Hydrogen power production has received a lot of interest recently. The benefit of hydrogen power generation is that it can provide energy without contributing to climate change by emitting carbon dioxide [7]. The proton exchange membrane fuel cell (PEMFC) stands out as a sustainable device for electricity generation with a wide range of applications, particularly in automotive [8] and material handling applications [9]. It also offers superior fuel

(hydrogen) economy, durability, and reliability compared to other power sources such as internal combustion engines and batteries [10]. However, renewable and sustainable energy sources, such as photovoltaic systems and fuel cells, necessitate a power conditioning system [11]. This is because the voltage produced by PEMFC is inherently low and must be elevated to a higher voltage level for various applications [12]. Typically, a DC/DC boost converter is employed for this purpose. Nevertheless, due to the influence of temperature fluctuations and gas pressure variations on PEMFC performance, it becomes imperative to develop a comprehensive control strategy to enhance performance and ensure efficient power conversion from the cell to the load.

### 1.2. State of the Art

Previous studies have achieved the main control objectives and thus have a massive impact on PEMFC control. For instance, the PI and PID controllers are considered the most popular controllers in the industry because of their simplicity and ease of implementation [13]. They are widely applied in control systems, including the control of pressure, temperature, humidity, etc. in PEMFC systems. Derbeli et al. [14] designed a PI controller, which was applied to a DC/DC boost converter linked to a PEMFC power system. Simulation results exhibit that the PI shows good performance in terms of rising time, overshoot, undershoot, and steady-state error. Silaa et al. [15] implemented an adaptive PID control strategy using stochastic gradient descent with momentum (SGDM) to control a DC/DC boost converter in order to achieve and obtain the required performance under a variety of disturbances. Simulation results show that the PIDSGDM controller can attain fast convergence and high robustness under extreme changes in temperature and load. According to [16,17], high-order sliding mode (HOSM) has the advantages of guaranteed stability, strong robustness, fast response speed, insensitivity to parameter perturbation, and ease of implementation. Considering the HOSM advantages over other control techniques, Russo et al. [18] introduced an innovative approach based on HOSM saturation for specific types of DC/DC converters. The objective of this approach is to ensure both the boundedness and smoothness of the duty cycle that feeds the pulse-width modulator (PWM) signal. The proposed control framework combines the bounded integral control (BIC) technique with a discontinuous HOSM control algorithm. Notably, the efficacy of the proposed approach was validated through numerical simulations. The results affirm the success of the proposed approach in stabilizing the output voltage of DC/DC converters while maintaining a smooth and bounded duty cycle behavior. Silaa et al. [19] controlled a DC/DC converter using a quasi-continuous high-order sliding mode control (QC-HOSM). The control strategy aims to hold the PEMFC power system to work at an adequate and efficient power point. Experimental results exhibit that the proposed controller can achieve high robustness, fast convergence, and a chattering decrease of more than 84%. Laghrouche et al. [20] introduced a controller for a PEMFC gas supply system. The controller was established using Lyapunov-based robust and adaptive HOSM. The performance of the proposed controller was evaluated through simulations, which showed that it has good performance in terms of fast convergence, tracking error, and robustness under stack current variation. Overall, this controller shows promise as a control technique for nonlinear systems with bounded uncertainty, such as the PEMFC gas supply system. Rakhtala et al. [21] designed an HOSM observer to provide the peroxygen ratio in an unmeasurable state by applying a second-order sliding mode to control the respiration of proton exchange membrane fuel cells (PEMFCs) through hypertension and suboptimal controllers. The results verify that the designed controller has better tracking performance. Moreover, the fuzzy logic controller (FLC) was adopted in several respects—this final option belongs to intelligent control—which is essentially a nonlinear form of control [22]. One of the most important features of FLC is the combination of systematic theory and practical application background. A dynamic model of the PEMFC temperature mechanism based on FLC has been established. Peng et al. [23] developed a two-dimensional incremental fuzzy logic controller (IFLC) to regulate the PEMFC temperature. This controller incorporated an integral

component based on a pre-established temperature model and empirical control guidelines. The outcomes of their research indicate that the model effectively replicates the dynamic behavior of the PEMFC. Moreover, when applied to the power system, the designed controller can actively maintain the temperature of the PEMFC within the desired operational range, showcasing impressive robustness in real-time temperature control. Furthermore, various control approaches have been incorporated into the FLC. For instance, Nguyen [24] proposed combining PI, SMC, and FLC approaches (FSMPIF) in order to improve ride comfort and enhance vehicle stability. Active suspension systems help reduce car vibrations from road interactions. Using this method and numerical modeling, car vibrations decreased by up to 12.59% in tests. Nguyen et al. [25] designed an adaptive FLC, SMC, and PID tuned by fuzzy logic (AFSPIDF). While PID suits linear systems, SMC can face chattering in nonlinear systems and the fuzzy algorithms can add control flexibility. With the SMC output informing one piece of fuzzy logic and another adjusting the PID parameters, this combined method showed, through numerical tests, a reduction in vehicle vibrations by up to 13.5%, increasing stability and comfort. According to the combined approach's results and performance, Wei et al. [26] introduced a fuzzy PID in the control of heat exchangers and used the Matlab FLC toolbox to design the fuzzy reasoning system. The fuzzy PID controller was designed in Simulink, and the step response curve of the system after the introduction of fuzzy control was obtained. The results indicate that, in comparison to traditional PID control, fuzzy PID control has a smaller overshoot, higher precision, and faster stabilization speed, which can achieve the expected constant temperature output, can effectively improve heat transfer rate, and can reduce energy loss. Li et al. [27] introduced a robust fuzzy SMC in order to enhance the PEMFC system performance by effectively regulating the air supply flow. Numerical simulations confirmed an elimination of jitter in comparison to traditional SMC. Notably, fuzzy SMC demonstrated superior performance over static feed-forward (sFF), PI, and PI/sFF controllers when controlling PEMFC air supply flow, resulting in an improvement in system efficiency. Liping et al. [28] established a mathematical model of PEMFC and designed an adaptive fuzzy controller (AFLC) for the constant powering of fuel cells. Experiments showed that the designed controller can realize the constant power output from a PEMFC. The neural network (NN) is a highly nonlinear system, and neural network control (NNC) is an intelligent control method which is widely used in control systems because of its self-organization, self-learning, and self-adaptive capabilities. It is suitable for control analysis of complex systems such as PEMFC. According to the humidity characteristics of a PEMFC, Yin et al. [29] developed a model for effectively managing the oxygen excess ratio (OER) in the PEMFC power system. The air supply system was integrated with a fuel cell stack voltage model based on physical laws and empirical data. A conventional PID controller was initially implemented to regulate the OER. Subsequently, the gain coefficients were fine-tuned using an integration of FLC inference and an NN algorithm into the conventional PID controller. The simulation results demonstrated that the dynamic responses of the fuel cells were significantly improved in both constant and variable OER controls, with small overshoots and settling times of less than 0.2 s. Li et al. [30] developed a temperature-based back propagation neural network (BP-NN) controller with strong control performance that does not depend on high model accuracy. Through simulations, it was demonstrated that the BP-NN temperature control system exhibits good robustness and temperature control performance. Vinu et al. [31] developed a voltage output feedback controller that employs the neural network feed-forward (NNFF) in combination with the harmony search algorithm (HSA) to regulate the output voltage of the PEMFC. A comparison was made between the suggested controller and the NNFF controller, demonstrating that the proposed controller achieves closer tracking of the reference voltage. To evaluate performance, the controller was assessed using the integral square error (ISE), integral absolute error (IAE), and integral time-weighted absolute error (ITAE). The results indicate that the proposed controller has the lowest systematic error value and superior overall performance. As the quest for precise PEMFC models persists, evolutionary optimization methods have emerged as indispensable tools for robust pa-

rameter estimation. Singla et al. [32] proposed the black widow optimization algorithm (BWOA) for this need. Herein, the algorithm's effectiveness is first validated through intricate benchmarking, showcasing its versatility in complex problem-solving scenarios. Subsequently, the BWOA algorithm is applied to extract critical parameters from PEMFC models across diverse operating temperatures. Comparative analysis ensues, contrasting parameter optimization outcomes achieved through BWOA against five alternative state-of-the-art algorithms: particle swarm optimization (PSO), multi-verse optimization (MVO), sine cosine algorithm (SCA), whale optimization algorithm (WOA), and grey wolf optimization (GWO). Through meticulous error analysis across two distinct PEMFC datasets, the research affirms the superior performance of the developed BWOA compared to the alternative optimization approaches. Additionally, non-parametric testing confirms the consistent supremacy of BWOA over the array of competing algorithms. To summarize, the study encompasses a comprehensive exploration of parameter optimization in PEMFCs, highlighting the significance of advanced optimization techniques. The novel BWOA, due to its efficacy in benchmarking and PEMFC parameter optimization, is poised to enhance fuel cell control and efficiency while exemplifying algorithmic robustness within the field.

Despite the significant progress achieved in the PEMFC control field, several limitations persist, such as the presence of uncertainties, control accuracy, robustness, and nonlinearities. However, many control methods have emerged that underscore the need for more advanced and robust strategies.

### 1.3. Contributions

The original contribution of this study lies in the development of a novel PI sliding mode controller-based super-twisting algorithm (PISMCSTA), which is designed to address the limitations of conventional control approaches in PEMFC systems. While existing methods have valuable attributes, such as simplicity, robustness, and adaptability, they often struggle to achieve an optimal balance between control accuracy, robustness, and the suppression of unwanted chattering effects in order to optimize the net output power of the PEMFC. This paper identifies a research gap that lies in the need for a comprehensive control solution that can simultaneously improve the accuracy, robustness, and stability of PEMFC systems under varying operating conditions.

### 1.4. Structure Overview

This article is divided into seven sections. Section 2 describes the PEM fuel cell power system modeling. Section 3 is devoted to the DC/DC boost converter. Section 4 is devoted to the control design methodology. Section 5 represents the BWOA optimization technique. Section 6 exhibits the results and Section 7 presents the conclusions.

### 2. PEM Fuel Cell Power System Modeling

A fuel cell-type PEM makes it possible to directly convert the chemical combustion energy of hydrogen into electrical energy. Hence, the products of this reaction are heat and water. The PEMFC core consists of three elements. The first includes two electrodes: an oxidizing anode that acts as an electron emitter and a reducing cathode that acts as an electron collector separated by an electrolyte. The latter has the properties of conducting ionized molecules (protons) directly from one electrode to another and of blocking electrons by forcing them to pass through the external circuit where their electromotive energy can be exploited [33]. The PEMFC work principle can be summarized in Figure 1. At the PEMFC core, which is an acidic solid membrane, two electrochemical reactions take place successively [34].

At the level of the anode, in the presence of platinum, a catalytic oxidation occurs and the hydrogen dissociates from its electrons [34]:

$$2H_2 \implies 4H^+ + 4e^- \tag{1}$$

At the level of the cathode, in the presence of platinum, a catalytic reduction occurs: the oxygen combines with the protons that have crossed the electrolyte membrane and the electrons arriving from the external circuit. The reaction produces heat and water [34]:

$$4H^+ \quad + \quad O_2 \quad + \quad 4e^- \quad \Longrightarrow \quad 2H_2O + Heat \tag{2}$$

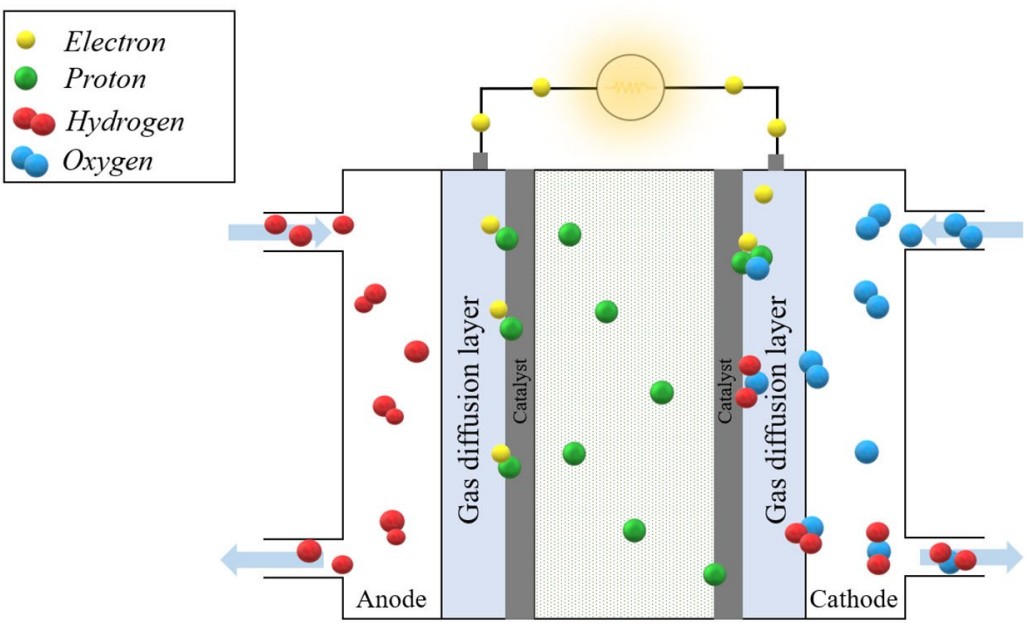

**Figure 1.** A cross-section of a PEMFC.

A single PEMFC voltage can be described as follows [34]:

$$V_{fc} = E_{Nernst} - E_{Act} - E_{Ohm} - E_{Con} \tag{3}$$

In Equation (3), the different terms $E_{Nernst}$, $E_{Act}$, $E_{Ohm}$, and $E_{Con}$ are the Nernst thermodynamic voltage, the activation polarization, the ohmic polarization, and the concentration polarization, respectively.

### 2.1. Nernst Voltage

The Nernst voltage represents the reversible thermodynamic voltage of the electrochemical reaction, which is given as follows [35]:

$$E_{Nernst} = 1.299 - 0.85 \cdot 10^{-3} \cdot (T_{FC} - 289.15) + 4.3085 \cdot 10^{-5} \cdot T_{FC}\left[ln(P_{H_2}) + \frac{1}{2} \cdot ln(P_{O_2})\right] \tag{4}$$

where $T_{FC}$ is the cell temperature, and $P_{O_2}$ and $P_{H_2}$ are the inlet oxygen and the hydrogen gas pressures, respectively.

### 2.2. Activation Polarization

The activation polarization occurs due to the electrochemical reactions which require a certain amount of energy to overcome the energy barrier for the electrochemical reaction to proceed [35]:

$$E_{act} = \zeta_1 + \zeta_2 \cdot T_{FC} + \zeta_3 \cdot T_{FC} \cdot ln(C_{O_2}) + \zeta_4 \cdot T_{FC} \cdot ln(i_{FC}) \tag{5}$$

where $i_{FC}$ is the operating current and the parameters $\zeta_{1,2,3,4}$ represent the parametric coefficients for each PEMFC model. The term $C_{O_2}$ is the oxygen concentration in the cathode catalytic interface (mol/cm$^3$) which is given as follows [35]:

$$C_{O_2} = \frac{P_{O_2}}{5.08 \cdot 10^6 \cdot e^{\left(\frac{-498}{T_{FC}}\right)}} \tag{6}$$

### 2.3. Ohmic Polarization

The ohmic polarization results from the electron transfer resistance across the collector plates and carbon electrodes, denoted $R_C$, and the proton movement resistance across the solid membrane, denoted $R_M$. The equivalent membrane resistance is given as follows [35]:

$$R_M = \frac{\sigma_M \cdot l}{A} \tag{7}$$

where $\sigma_M$ is the specific resistivity of the membrane (Ohm.m), $A$ is the active area of the cell (cm$^2$), and $l$ is the thickness of the membrane in cm.

Also, the specific resistivity of the PEMFC membrane can be given as follows [35]:

$$\sigma_M = \frac{181.6[1 + 0.03(\frac{i_{FC}}{A}) + 0.062(\frac{T_{FC}}{303})^2 \cdot (\frac{i_{FC}}{A})^{2.5}}{[\gamma - 0.634 - 3(\frac{i_{FC}}{A})] \cdot \exp[4.18(T_{FC} - 303)/T_{FC}]} \tag{8}$$

Therefore, the ohmic polarization is given as follows [35]:

$$E_{Ohm} = i_{FC} \cdot (R_M + R_C) \tag{9}$$

### 2.4. Concentration Polarization

The concentration polarization occurs due to the diminution in the density of the reacting materials. This polarization can be calculated using Equation (10) [35]:

$$E_{con} = \psi \cdot ln\left(1 - \frac{J}{J_{max}}\right) \tag{10}$$

where $\psi$ is a parametric coefficient that depends on the cell, $J$ represents the cell current density in A $\cdot$ cm$^2$, and $J_M$ is the maximum current density.

Therefore, using Equations (4), (5), (9) and (10), the overall PEMFC stack output power is given as follows [35]:

$$P_{FC} = V_{fc} \cdot i_{FC} \cdot N_{Cells} \tag{11}$$

Table 1 presents the different parametric coefficients of the PEMFC used in the simulation.

**Table 1.** PEM fuel cell parameters.

| Parameter | Value |
|---|---|
| A | 162 cm$^2$ |
| $\gamma$ | 23 |
| $l$ | $175 \times 10^{-6}$ cm |
| $\psi$ | 0.1 V |
| $R_C$ | 0.0003 |
| $J_M$ | 0.062 A $\cdot$ cm$^{-1}$ |
| $N_{Cells}$ | 10 |
| $\zeta_1$ | $-0.9514$ V |
| $\zeta_2$ | $-0.00312$ V/K |
| $\zeta_3$ | $-7.4 \times 10^{-5}$ V/K |
| $\zeta_4$ | $1.87 \times 10^{-4}$ V/K |

## 3. DC/DC Boost Converter Linked to PEMFC

In many applications related to clean energy, fuel cells are combined with power converters which produce an efficient path from the cell stack to the load, which also delivers a regulated output voltage [36]. Assorted configurations of converters are available; the most simple are the buck, the boost, and the buck–boost converters. They give a controlled output voltage based on a PWM signal generated by a discrete device [36].

The selection of the type of converter depends on the end-user voltage required. In this case, the intention was to increase the value; thus, a boost type was found to be optimal. Due to the low PEMFC output voltage, a high step-up power converter was used. This converter plays a pivotal role in elevating and stabilizing the output PEMFC voltage and ensuring it functions at an optimal power point while delivering a suitable direct current (DC). On the other hand, the PEMFC output characteristics are influenced by changes in several parameters, such as the cell temperature, the oxygen and hydrogen partial pressures, and the load demands [36]. Therefore, a control algorithm must be established for optimal and proper operation.

Figure 2 shows the equivalent electric circuit, which consists of an inductor (L), a diode (D), a switching device ($S_1$), a capacitor (C), and a load (R). The device ($S_1$) generates a PWM signal that switches between ON and OFF to adjust the voltage.

$$V_o = \left( \frac{1}{1-d} \right) \cdot V_s \tag{12}$$

Equation (12) is related to the output voltage [37], where $V_o$, $d$, and $V_s$ are the output voltage, the duty cycle, and the voltage generated by the PEMFC. This implies that, when $d$ increases, the output voltage follows the same trend. The mechanism of the boost converter is as follows. As the controller sends a PWM signal to the switching element, the states of the circuit shift in two different configurations [38]:

- ($S_1$) is ON: The inductor current will increase linearly until it reaches a peak value of current and, at this point, the voltage around the inductor will be equal to the input voltage source: $V_L = V_s$. In this step, the current in the inductor $i_L$ and the output voltage $V_o$ are dependent on the following dynamic (13):

$$\begin{cases} \frac{di_L}{dt} = \frac{1}{L}(V_s) \\[2mm] \frac{dV_o}{dt} = \frac{1}{RC}(-V_o) \end{cases} \tag{13}$$

- ($S_1$) is OFF: In this case, the inductor current $i_L$ gets discharged into capacity C; at the end of this action, the voltage in the inductor will be $V_L = V_s - V_o$. At this point, the system in (13) shifts into the following (i.e., (14)):

$$\begin{cases} \frac{di_L}{dt} = \frac{1}{L}(V_s - V_o) \\[2mm] \frac{dV_o}{dt} = \frac{1}{C}(i_L - i_o) \end{cases} \tag{14}$$

The previous equations can yield (15), which is the state-space equation that represents the boost converter dynamics [38]:

$$\begin{cases} \begin{bmatrix} \frac{di_L}{dt} \\[2mm] \frac{dV_o}{dt} \end{bmatrix} = \begin{bmatrix} 0 & \frac{-(1-u)}{L} \\[2mm] \frac{(1-u)}{C} & -\frac{1}{RC} \end{bmatrix} \cdot \begin{bmatrix} i_L \\[2mm] V_o \end{bmatrix} + \begin{bmatrix} \frac{1}{L} \\[2mm] 0 \end{bmatrix} V_s \\[6mm] y = \begin{bmatrix} 0 & 1 \end{bmatrix} \cdot \begin{bmatrix} i_L \\[2mm] V_o \end{bmatrix} \end{cases} \tag{15}$$

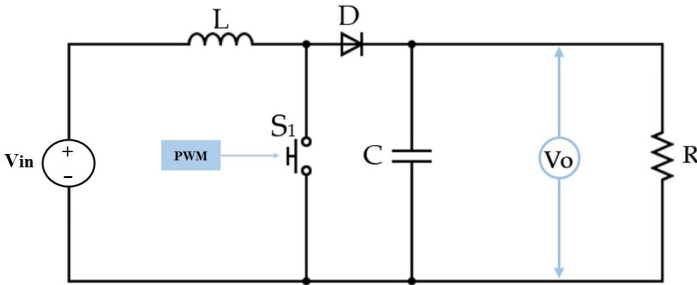

**Figure 2.** DC/DC boost converter circuit.

The parameters of the DC/DC boost converter used in the simulation are presented in Table 2.

**Table 2.** DC/DC boost converter parameters.

| Parameter | Value |
|---|---|
| Inductance | 69 mH |
| Capacitor | 1.5 mF |
| Maximum switching frequency | 10 kHz |
| Maximum input voltage | 25 V |
| Maximum input current | 15 A |
| Maximum output voltage | 80 V |
| Maximum output current | 2 A |

## 4. Control Design

In this research, we compared two types of robust controllers: a conventional super-twisting algorithm (STA) and the novel PI sliding mode controller-based super-twisting algorithm (PISMCSTA). These two approaches were designed so that the PEMFC could follow a reference current named $i_{ref}$ with fast-tracking error compensation. The simulation blocks of the closed-loop system consisted of a PEMFC stack, a DC/DC converter-type boost, the BWOA technique, a PISMCSTA controller, the *P&O* MPPT technique, and, finally, variable load, as shown in Figure 3.

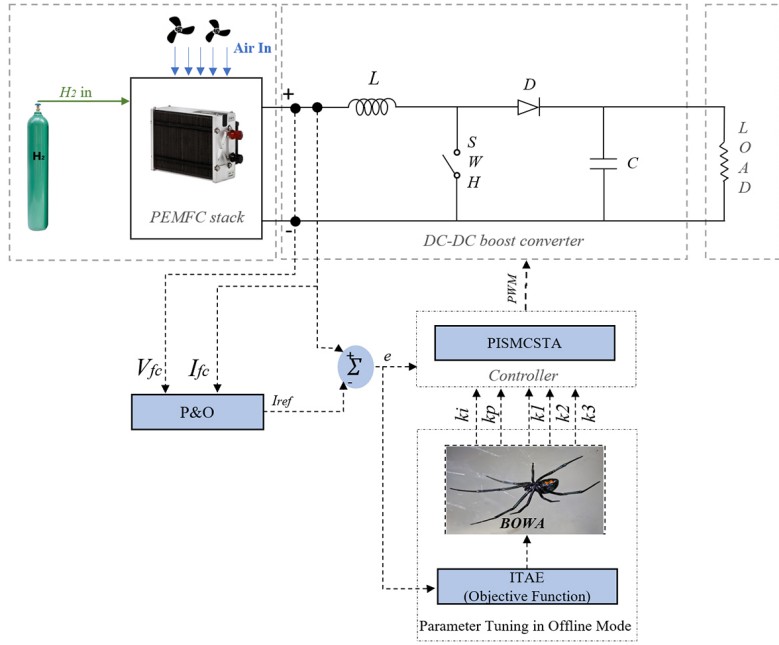

**Figure 3.** Matlab Simulink closed-loop system.

### 4.1. Reference Current Based P&O MPPT

The perturb and observe (*P&O*) algorithm is a popular maximum power point-tracking (MPPT) technique that is used in order to track the maximum power point (MPP) of a renewable energy system, such as a photovoltaic (PV) system or a fuel cell, like PEM-FCs [39]. In the context of the PEMFC power system, the *P&O* algorithm can be used to adjust the reference current ($I_{ref}$) to ensure that the PEMFC operates at its maximum output power, which the algorithm attempts to reach. The basic principle of the *P&O* algorithm involves a series of iterative steps to optimize power extraction from the PEMFC. It begins with an initial reference current value and measures the output power of the PEMFC system by monitoring the voltage and current on the PEMFC stack. The algorithm then slightly increases the reference current ($\Delta I_{ref}$) and observes the corresponding change in power output. Depending on the observed power variation, the algorithm either continues in the same direction with the new reference current (if the power output increases), or changes the direction of perturbation (if the power output decreases). These steps are repeated until the MPP is reached, identified by the power output starting to decrease [39]. Figure 4 shows the reference current *P&O* algorithm diagram.

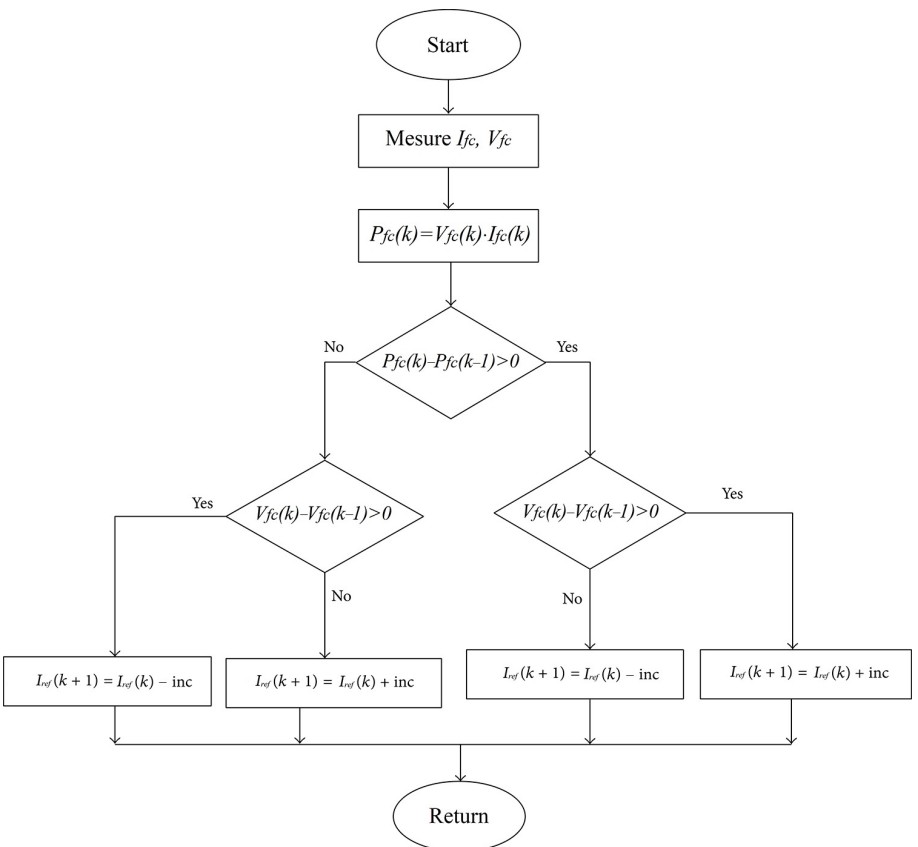

**Figure 4.** The *P&O* reference current flowchart.

### 4.2. Super-Twisting Algorithm

As for any *SMC* type, the control law expression is composed of an equivalent term (or $u_{eq}$), whose intention is to preserve the system at the surface, and a switching term (or $u_{sw}$), which is used to bring the system to the sliding surface [40]. The latter also takes into account that preservation implies dealing with disturbances, unknown dynamics, etc. As a mathematical expression, this can be defined as the following expression, which was used for further design of the controllers [41]:

$$u = u_{eq} + u_{sw} \tag{16}$$

As previously mentioned, the main aim of the proposed control scheme is for the PEMFC generated current $i_L$ to track the reference current $i_{ref}$, which provides the maximum power. Therefore, the error signal used for the development of the proposed controllers is defined as follows:

$$e = i_L - i_{ref} \tag{17}$$

The first step in the design of the *STA* is to choose an adequate surface; in our case, we followed the suggestions of the authors from [41]. The general formula of the sliding surface is given as follows:

$$s = \left(\frac{d}{dt} + \lambda\right)^{r-1} \int_0^t e\, dt \tag{18}$$

Expression (18) has the constants $r$ and $\lambda$ which are, respectively, the relative degree of the system and a positive constant that is associated with the bandwidth of the control to be designed. Therefore, by setting the relative degree $r$ equal to 2, the sliding surface for the first proposed controller (*STA*) can be expressed by Equation (19):

$$s_1 = e + \lambda \int_0^t e\, dt \tag{19}$$

In the context of the proposed control strategies, the dynamics of the reference current ($i_{ref}$) change over the time due to the *P&O* MPPT technique. However, after a closer examination of the system behavior and its impact on the control strategy, the decision was made to simplify without sacrificing the control strategies by omitting the derivative of the reference current. Additionally, the analysis determined that the rate of change of $i_{ref}$ either has minimal influence on the overall system dynamics or is relatively small. Essentially, when compared to more significant system variables such as inductor current ($i_L$) and the control inputs, $i_{ref}$ can be treated as a slowly varying or quasi-constant parameter. Therefore, by using (15), (17), and (19), the surface derivative can be expressed as the following:

$$\dot{s}_1 = \frac{V_o}{L}u - \frac{V_o}{L} + \frac{V_s}{L} + \lambda \cdot e \tag{20}$$

As previously enacted, (16) refers to two terms that are related to each controller. In order to provide differentiation in both designs, we define (21), which establishes the control signal $u_c$ for the conventional *STA*:

$$u_c = u_{eq_c}(t) + u_{sw_c}(t) \tag{21}$$

According to the authors from [41], the equivalent control term can be deduced from the statement $\dot{s}_1 = 0$. The usage of (20) allows one to obtain the following expression:

$$u_{eq_c}(t) = 1 - \frac{V_s}{V_o} - \frac{\lambda \cdot e \cdot L}{V_o} \tag{22}$$

On the other hand, the switching term for this case is defined by (23) (based on reference [41]):

$$u_{sw_c}(t) = -\frac{L}{V_o}k_1 \cdot |s_1|^{0.5} \cdot sign(s_1) - \frac{L}{V_o}k_2 \int sign(s_1)dt \tag{23}$$

such that $k_{1,2} > 0$, and its choice is very influential because a small value can increase the response time; oppositely, for a higher value, strong oscillations can occur. These effects can excite neglected dynamics (chattering phenomenon) or even deteriorate the hardware [42].

### 4.3. Novel PI Sliding Mode Super-Twisting Controller

This work proposes super-twisting combined with sliding mode control, as well as the *PI* surface being the mathematical expression from [43,44]. The *PI* controller time domain can be defined as follows [44]:

$$U_{PI}(t) = K_P e(t) + K_I \int e(t)dt \tag{24}$$

Both controller *PI* and *STA* have advantages and disadvantages. However, the main advantage of the *PI* is to provide simple and good performance, but presents a lack of robustness under system uncertainties. On the other hand, *STA* is robust and stable against perturbations and can overcome the chattering effect that usually appears in the *SMC*. Thus, by combining both *PI* control and *STA*, we can obtain a suitable controller. The sliding surface is based on the *PI* controller, which is described as follows:

$$s_2 = K_P e(t) + K_I \int e(t)dt \tag{25}$$

Differentiating Equation (25) with respect to time yields

$$\dot{s}_2 = K_P \dot{e} + K_I e \tag{26}$$

Substituting Equations (15) and (17) into the previous equation, we find

$$\dot{s}_2 = K_P\left(\frac{V_o}{L}u_s - \frac{V_o}{L} + \frac{V_s}{L}\right) + K_I e \tag{27}$$

The established *PISMCSTA* controller signal is defined as $u_s$, which is given as follows:

$$u_s = u_{eq_s}(t) + u_{sw_s}(t) \tag{28}$$

By forcing $\dot{s}_2 = 0$, the equivalent control can, therefore, be deduced as follows:

$$u_{eq_s}(t) = 1 - \frac{V_s}{V_o} - \frac{K_I e L}{K_P V_o} \tag{29}$$

The proposed novel switching control-based *PI* sliding mode super-twisting algorithm is given as follows:

$$u_{sw_s}(t) = -\frac{L}{V_o K_P} sign(s_2)\left(k_1 + k_2 \cdot |s_2|^{0.5}\right) - \frac{L}{V_o K_P} k_3 \int sign(s_2)dt \tag{30}$$

where $k_{1,2,3}$ are positive constants.

### 4.4. Stability Proof of STA and PISMCSTA

In order to prove the stability of the *STA*, a candidate Lyapunov function can be defined as follows:

$$V_1 = \frac{1}{2}s_1^2 \tag{31}$$

To ensure that the proposed Lyapunov function $V_1$ converges to zero in finite time, its derivative $\dot{V}_1$ must be semi-negative. By using Equations (19), (20), (22) and (23), differentiating Equation (31) with respect to time yields the following:

$$
\begin{aligned}
\dot{V}_1 &= s_1 \dot{s}_1 \\
&= s_1 \left[ \frac{V_o}{L} u_c - \frac{V_o}{L} + \frac{V_s}{L} + \lambda e \right] \\
&= s_1 \left[ \frac{1}{L}(V_s - V_o) + \frac{V_o}{L}(u_{eq_c} + u_{sw_c}) + \lambda e \right] \\
&= s_1 \left[ \frac{1}{L}(V_s - V_o) + \frac{V_o}{L} u_{eq_c} + \frac{V_o}{L} u_{sw_c} + \lambda e \right] \\
&= s_1 \left[ \frac{1}{L}(V_s - V_o) + \frac{V_o}{L}\left( -\frac{1}{V_o}(V_s + \lambda L e - V_o) \right) \right] \\
&\quad + s_1 \left[ \frac{V_o}{L}\left( -\frac{L}{V_o} k_1 \cdot |s_1|^{0.5} \cdot sign(s_1) - \frac{L}{V_o} k_2 \int sign(s_1)dt \right) + \lambda e \right] \\
&= s_1 \left[ -k_1 \cdot |s_1|^{0.5} \cdot sign(s_1) - \int k_2 \cdot sign(s_1)dt \right] \\
&\leq 0
\end{aligned}
\tag{32}
$$

Stability is achieved when both $k_{1,2}$ are positive and greater than 0. Consequently, according to Lyapunov theory, the PEMFC power system is stable.

Finally, we obtain the *PISMCSTA* stability proof in the same manner, by replacing $s_2$ as follows:

$$
V_2 = \frac{1}{2} s_2^2
\tag{33}
$$

To ensure that the proposed Lyapunov function $V_2$ converges to zero in finite time, its derivative $\dot{V}_2$ must be semi-negative. By using Equations (25), (26), (29), and (30), differentiating Equation (33) with respect to time yield

$$
\begin{aligned}
\dot{V}_2 &= s_2 \dot{s}_2 \\
&= s_2 \left[ K_P\left( \frac{V_o}{L} u_s - \frac{V_o}{L} + \frac{V_s}{L} \right) + K_I e \right] \\
&= s_2 \left[ K_P\left( \frac{V_o}{L}(u_{eq_s} + u_{sw_s}) - \frac{V_o}{L} + \frac{V_s}{L} \right) + K_I e \right] \\
&= s_2 \left[ \frac{K_P V_o}{L} u_{eq_s} + \frac{K_P V_o}{L} u_{sw_s} - \frac{K_P V_o}{L} + \frac{K_P V_s}{L} + K_I e \right] \\
&= s_2 \left[ \frac{K_P V_o}{L}\left(1 - \frac{V_s}{V_o} - \frac{K_I e L}{K_P V_o}\right) - \frac{K_P V_o}{L} + \frac{K_P V_s}{L} + K_I e \right] \\
&\quad + s_2 \left[ \frac{K_P V_o}{L}\left( -\frac{L}{V_o K_P} sign(s_2)\left( k_1 + k_2 \cdot |s_2|^{0.5} \right) - \frac{L}{V_o K_P} k_3 \int sign(s_2)dt \right) \right] \\
&= s_2 \left[ -sign(s_2)\left( k_1 + k_2 \cdot |s_2|^{0.5} \right) - k_3 \int sign(s_2)dt \right] \\
&\leq 0
\end{aligned}
\tag{34}
$$

When $k_1$, $k_2$, and $k_3$ are all greater than 0, stability is achieved [45]. Consequently, according to the Lyapunov theory, the PEMFC power system is stable.

## 5. Optimization Using the Black Widow Algorithm

The black widow optimization algorithm (BWOA) was proposed by Hayyolalam et al. [46] in 2020, inspired by the unique mating behavior of black widow spiders. The algorithm simulates the life cycle of black widow spiders. Each solution (or potential solution) of the BWOA is a black widow spider whose length is equal to the dimension of the search space. The BWOA includes five stages: population initialization, reproduction, cannibalism, mutation, and population update. In addition to the initial population stage, the other

four stages need to be iterated until the end conditions are met, and the black widow with the best fitness is returned.

### 5.1. Initializing the Population

The initialization of the black widow spider population is consistent with other intelligent optimization algorithms, and the population is randomly initialized within the boundary range. The black widow spiders can be regarded as a one-dimensional array as follows [46]:

$$Widow = [x_1, x_2 \ldots \ldots x_{N_{var}}] \tag{35}$$

where $N_{var}$ is an optimized dimension, and, when initialized, the value of each dimension is a random floating-point number. Each black widow has fitness and uses a fitness function to calculate the fitness of the black widow [46]:

$$Fitness = f(widow) = f(x_1, x_2 \ldots \ldots x_{N_{var}}) \tag{36}$$

when initializing the population, it is necessary to generate only $N_{pop}$ (population size) black widows, obtaining $N_{pop} \times N_{var}$ to generate a black widow matrix.

### 5.2. Reproduction

The reproductive portion is defined as the global search phase. First, the population is sorted according to the fitness, and the black widows participating in reproduction in the population are calculated based on the procreating rate (PR). Then, a pair of parents (male and female black widows) are randomly selected for mating and reproduction. In nature, each pair of black widows reproduces on their own web, separate from other black widows, producing about 1000 eggs each time and protecting them with a sac, as shown in Figure 5, but only the smaller spiders with better fitness survive. Hence, in the black widow algorithm, each pair of parents uses an array $\beta$ that simulates the reproductive process [46]:

$$\begin{cases} y_1 = \beta x_1 + (1 - \beta)x_2 \\ y_2 = \beta x_2 + (1 - \beta)x_1 \end{cases} \tag{37}$$

where $x_1$ and $x_2$ are parents, while $y_1$ and $y_2$ are offspring. The process is repeated $N_{var}/2$ times.

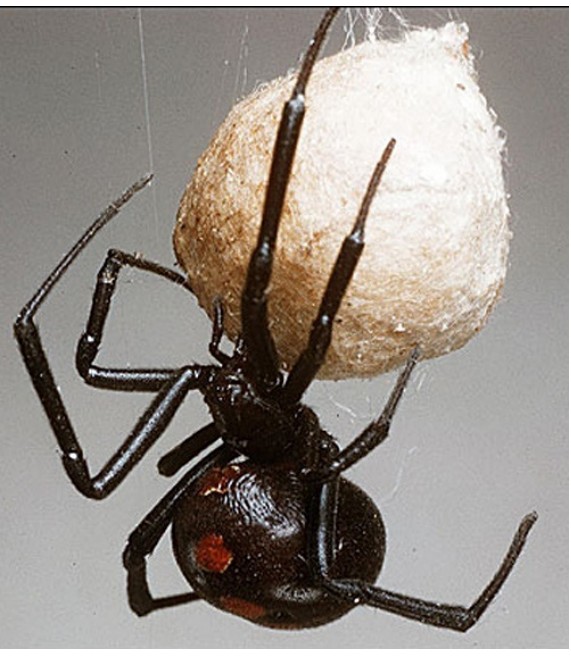

**Figure 5.** Female black widow with her egg sac.

### *5.3. Cannibalism*

Cannibalism is when the fittest spiders eat the poorly fit ones. There are three types of cannibalism in black widow spiders: sexual cannibalism, sibling cannibalism, and child cannibalism. Sexual cannibalism means that female black widows will eat male black widows during or after mating, and males and females can be distinguished according to their fitness. Spiders with good fitness are females, and spiders with poor fitness are males; sibling cannibalism occurs on the mother spider's web, and the young spiders live on the mother spider's web for about a week after hatching. During this period, sibling cannibalism will occur; cannibalism between the child and the mother refers to the event in which the young spider eats the mother spider under certain circumstances. Sexual cannibalism is achieved by destroying the father, and some children are destroyed according to the cannibalism rate (CR) to achieve the purpose of sibling cannibalism. The use of the fitness value has a strong utility in determining how the spiders are strong.

### *5.4. Mutation*

The mutation is a local search phase. At this stage, the BWOA randomly selects multiple black widows according to the mutation rate (MR), and each black widow randomly exchanges two values in an array.

### *5.5. Update Population*

After one iteration, the black widows retained in the cannibalism stage and the black widows obtained in the mutation stage are used as the initial population of the next iteration.

### *5.6. Stop Conditions*

Three stopping conditions can be considered: the maximum number of iterations is set in advance; the optimal black widow does not change anymore; and a preset level of accuracy is reached. The BWOA flowchart is shown in Figure 6.

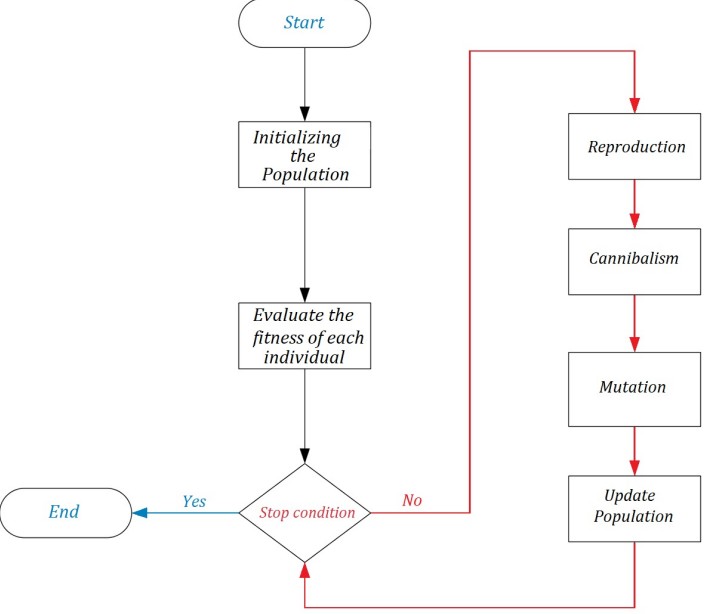

**Figure 6.** The BWOA flowchart [46].

In this simulation, the BWOA is used in order to tune the PISMCSTA and STA controllers parameters. $I_{ref}$ is the reference current extracted by the *P&O* technique. $I_L$ is the current produced by the PEMFC power system. *e* is error current, which denotes the difference between $I_L$ and $I_{ref}$. During each iteration of the BWOA, population values of

$K_{p,i,1,2,3}$ for the PISMCSTA and $K_{1,2}$ for the STA are generated, which are then substituted into the objective function ITAE [47]. PISMCSTA and STA controllers take $e$ as the input signal and then produce the corresponding control signal. The process is repeated until the error signal approaches zero. Finally, the best values of the parameters are identified and used to design the optimum PISMCSTA and STA controllers.

The selected values for population size, the maximum number of iterations, procreating rate (PR), cannibalism rate (CR), and mutation rate (PM) were taken as 40, 100, 0.6, 0.44, and 0.4, respectively. The variation ranges of the decision variables used in the simulation are given in Table 3.

**Table 3.** BWOA upper and lower bounds.

| Algorithm | Range | $\lambda$ | $K_p$ | $K_i$ | $K_1$ | $K_2$ | $K_3$ |
|---|---|---|---|---|---|---|---|
| BWOA | Min | $fixed = 0.5$ | 0 | 0 | 0 | 0 | 0 |
| | Max | $fixed = 0.5$ | 1 | 1 | 10 | 10 | 10 |

## 6. Results

The primary objective of this research is to implement a PI sliding mode controller-based super-twisting algorithm that is then applied to a DC/DC step-up converter, in order to maintain the performance of the PEMFC power system at a reference current ($I_{ref}$) produced by the $P\&O$ technique, which searches the reference current for the maximum power point. The oxygen gas pressure during this simulation remained constant at 1 bar throughout. In addition, a wide range of hydrogen gas pressures, temperatures and variable loads were used (Figure 7) in order to verify the control technique's capacity to cope with a wide range of conditions.

Table 4 lays out all of the necessary parameters for the implemented controllers that were used in this investigation. Thereafter, the convergence rate for BWOA objective functions is shown in Figure 8.

**Table 4.** Controllers obtained parameters.

| Controller | $\lambda$ | $K_p$ | $K_i$ | $K_1$ | $K_2$ | $K_3$ |
|---|---|---|---|---|---|---|
| STA | 0.5 | 0 | 0 | 0.2610 | 6.5047 | — |
| PISMCSTA | 0.5 | 0.0172 | 0.5049 | 0.1148 | 0.2642 | 7.8458 |

Figure 9a–d shows the PEMFC P–I and V–I polarization curves under different temperatures and hydrogen gas pressures. According to these figures, it is noticed that the output power of the PEMFC is influenced by the temperature and the hydrogen gas pressures. From these figures, it can be observed that, when both temperature and hydrogen gas pressure increase, the PEMFC attains a heightened capacity for power generation. This revelation underscores the significance of temperature and hydrogen gas pressure regulation as levers to change the PEMFC energy conversion capabilities. Increasing the temperature generally leads to improved electrochemical reaction kinetics, resulting in enhanced fuel cell performance. At higher temperatures, reactants diffuse more rapidly, reducing activation losses and internal resistances, thus improving the overall current output. However, excessively high temperatures can lead to material degradation, reduced membrane efficiency, and potential thermal-related issues. Therefore, maintaining an optimal temperature range is essential to balance improved performance with the durability and reliability of the PEMFC power system. On the other hand, hydrogen gas pressure affects the reactant supply and transport within the PEMFC. The higher hydrogen gas pressure typically results in better reactant delivery to the electrodes, facilitating more efficient electrochemical reactions. This leads to increased power output and improved performance.

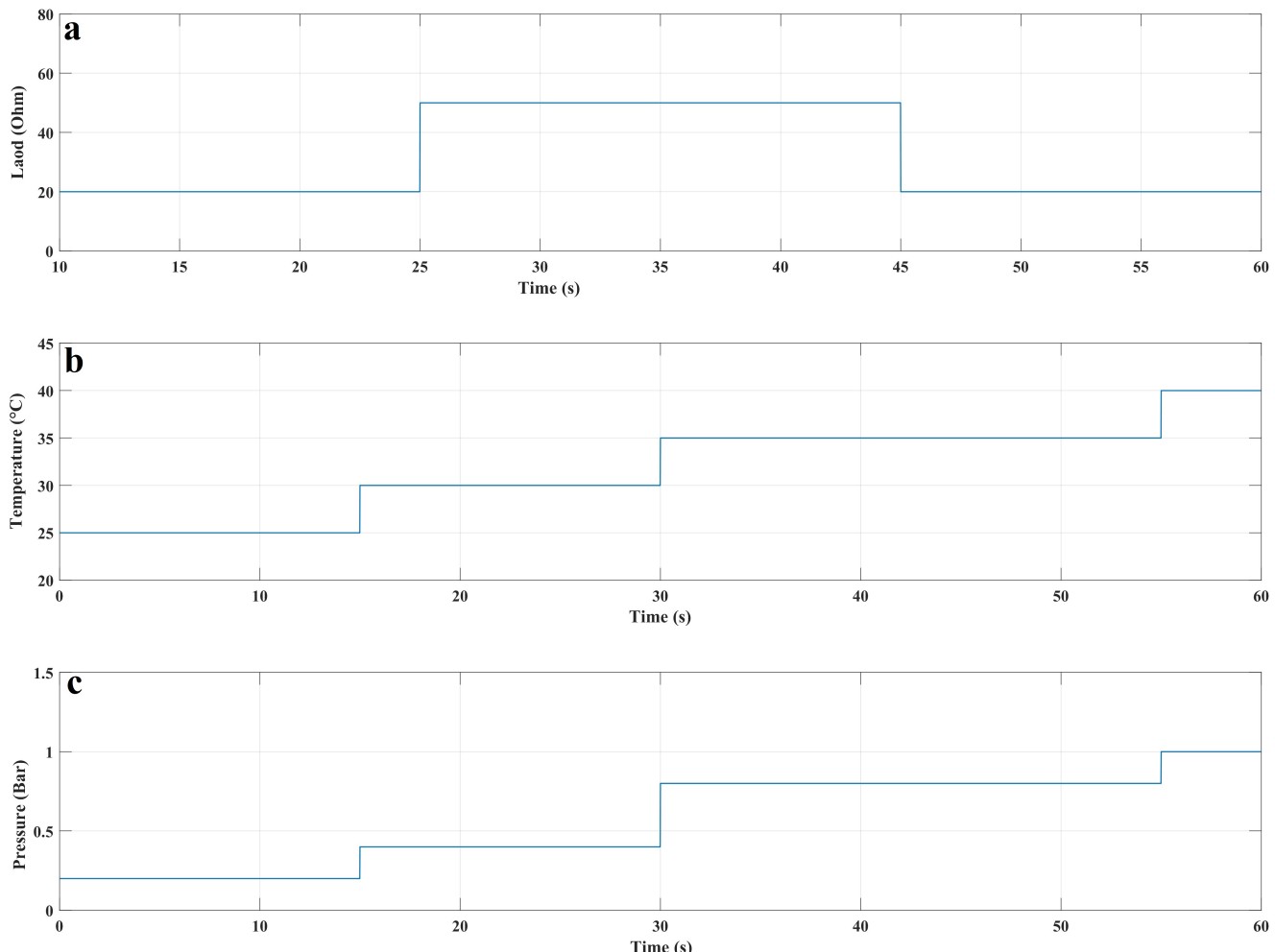

**Figure 7.** (**a**) Resistance; (**b**) temperature; (**c**) hydrogen gas pressure.

Figure 10a exhibits the demeanor of the PEMFC stack output voltage under the PISMCSTA and STA. According to this figure, it is noticeable, for the second time, that the PISMCSTA has faster convergence than the STA. On the other hand, at $t = 25$ s, the PISMCSTA shows an overshoot equal to 0.0374 V. Additionally, at $t = 45$ s, it shows an undershoot equal to 0.06 V. Going forward, from the period $t = 30$ s to $t = 45$ s, a voltage oscillation equal to 0.005 V and 0.014 V for PISMCSTA and STA, respectively, is observed; this value is 64% higher compared to the STA. Figure 10b shows the PEMFC stack output power under the PISMCSTA and STA. An analysis of the overshoot is presented; at $t = 30$ s, the PISMCSTA and STA controllers each have an overshoot, which is equal to 0.0449 W and 0.085 W, respectively. Moreover, at $t = 45$ s, the PISMCSTA and STA controllers each have an overshoot, which is equal to 0.18 W and 0.0534 W, respectively. On the other hand, at $t = 25$ s, the PISMCSTA shows an undershoot equal to 0.094 W. However, these overshoots and undershoots appear over short periods of time equal to 0.07 s. Going forward, from the period $t = 30$ s to $t = 45$ s, a power magnitude equal to 0.0162 W and 0.047 W for each PISMCSTA and STA, respectively, is observed; this is 66% higher compared to STA.

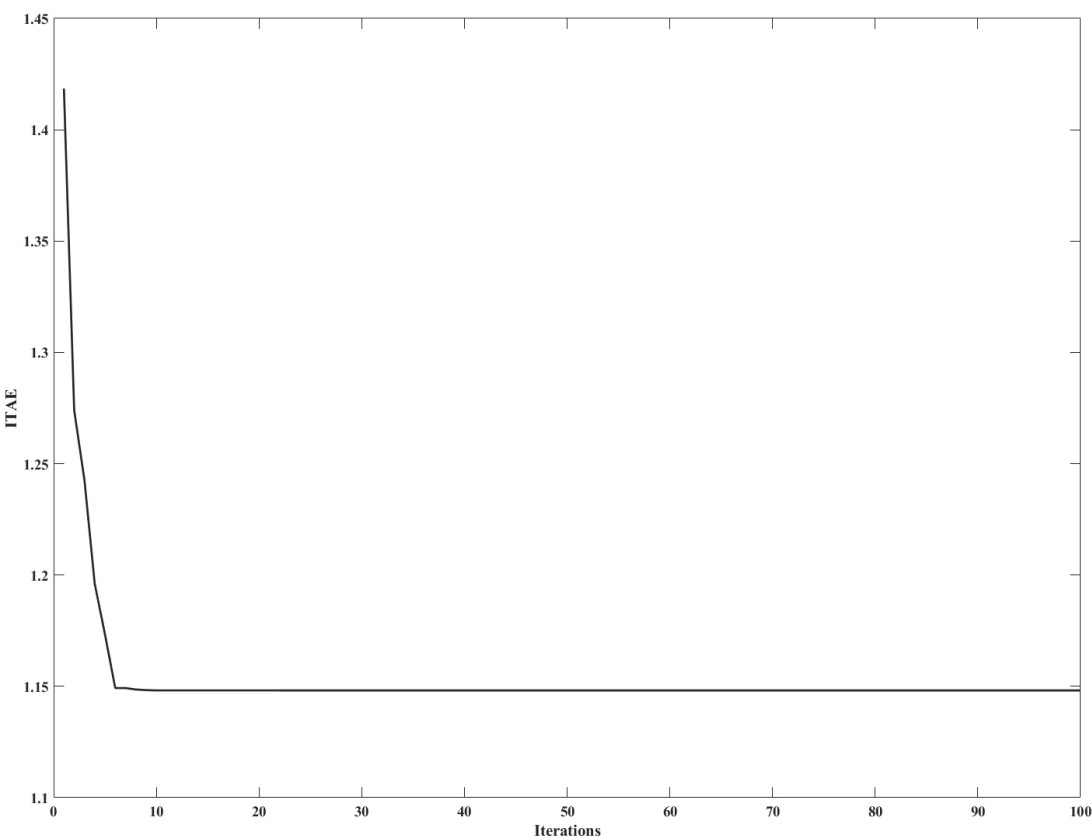

**Figure 8.** The convergence rate for BWOA objective function (ITAE).

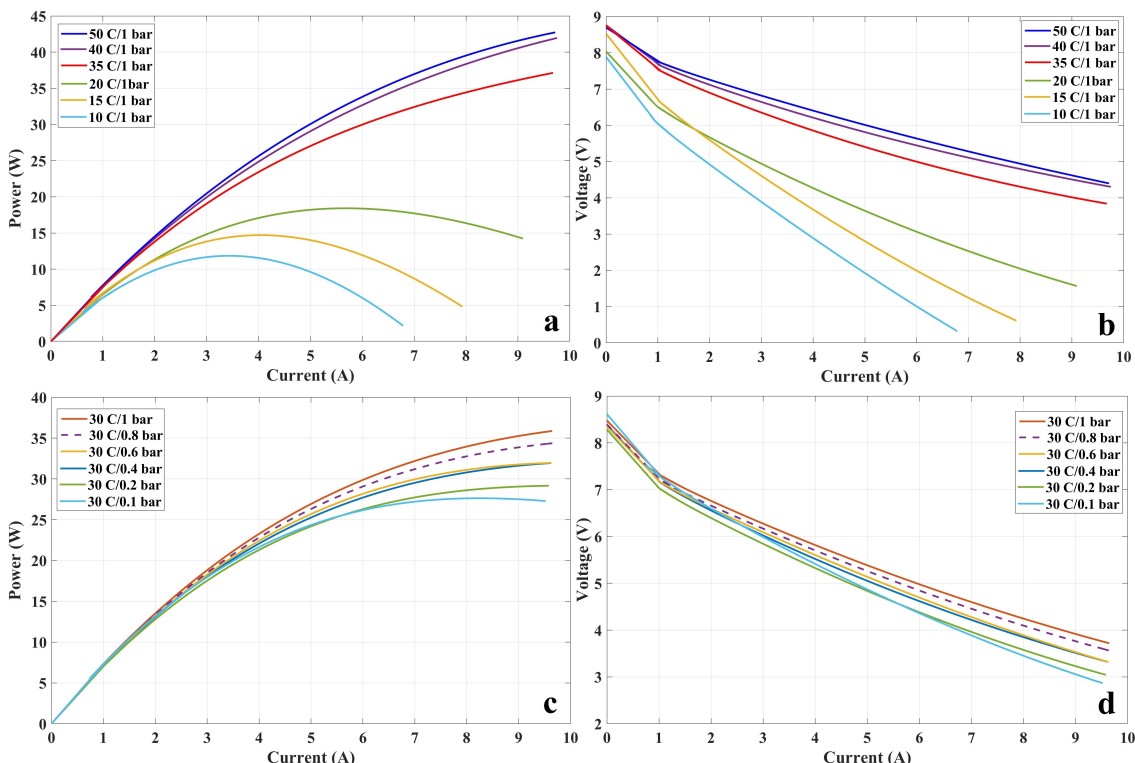

**Figure 9.** PEMFC polarization curves; (**a**) P–I curve under different temperature and fixed pressure; (**b**) V–I curve under different temperature and fixed pressure; (**c**) P–I curve under different pressure and fixed temperature; (**d**) V–I curve under different pressure and fixed temperature.

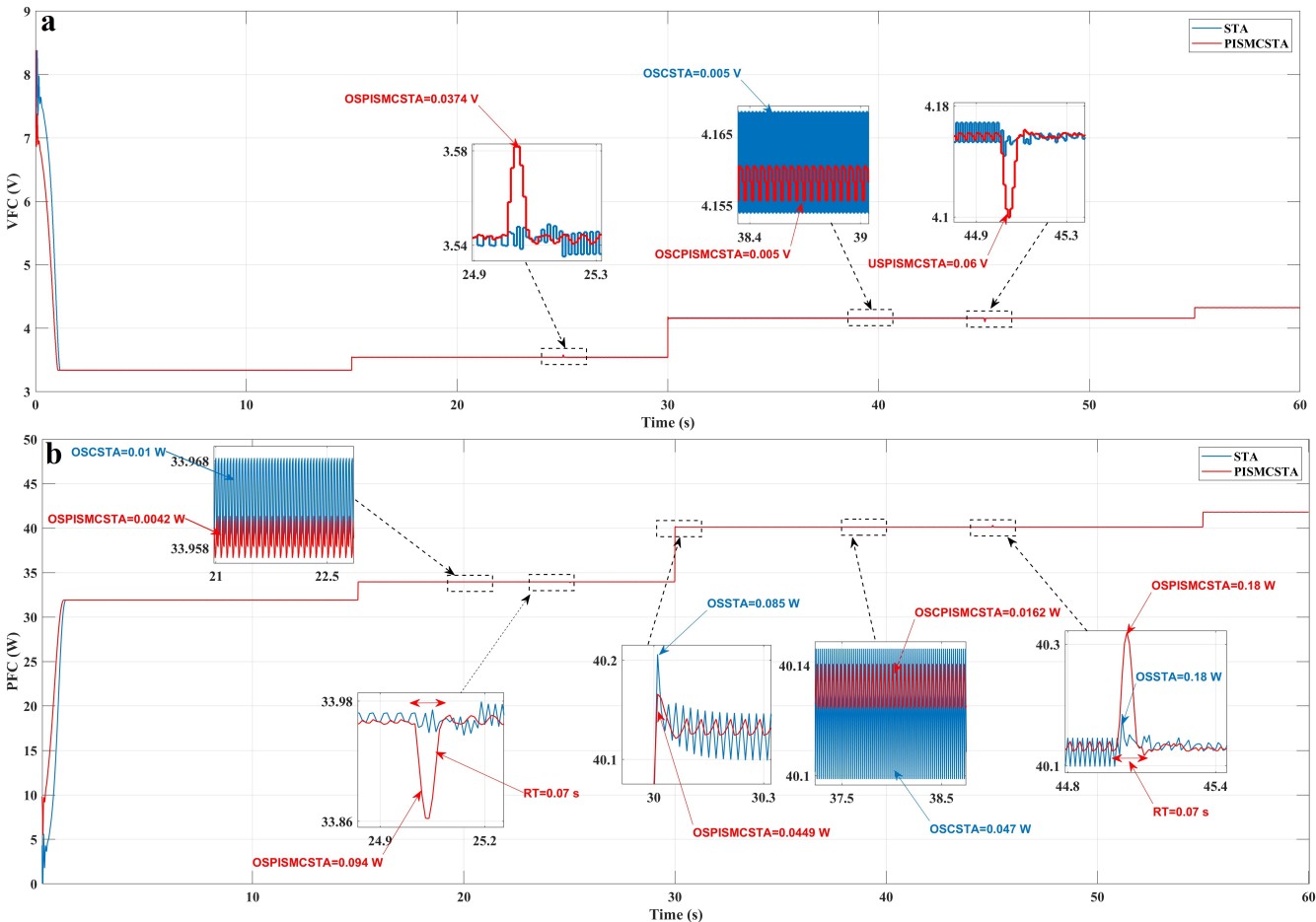

**Figure 10.** PEMFC stack output signals: (**a**) voltage; (**b**) power.

Figure 11a exhibits the duty cycle signal for both STA and PISMCSTA controllers. It is noticed that a soft and smooth rise to the desired reference value is exhibited by both controllers. However, it is evident that the drawbacks of the conventional STA are effectively overcome by the proposed PISMCSTA controller through the reduction of its chattering phenomenon. Figure 11b shows the performance of the PEMFC stack output current under two different control strategies, PISMCSTA and STA, when changes in temperature, hydrogen gas pressures, and load are subjected. Also, it is clearly seen that both STA and PISMCSTA controllers exhibit good performance: a gradual and smooth transition towards achieving the desired reference value of $I_{ref}$. However, it can be observed that this reference current is not fixed; it dynamically adapts to changes in operating condition, such as temperature and hydrogen gas pressure, in order to track the MPP in each operating condition in which the PEMFC is currently working. On the other hand, it is seen that the PISMCSTA has faster convergence than the other controller. Going forward to $t = 15$ s and $t = 55$ s, the PISMCSTA and STA have almost the same overshoot, at $t = 25$ s. Therefore, at $t = 45$ s, the PISMCSTA shows an overshoot equal to 0.18 A. On the other hand, at $t = 25$ s, the PISMCSTA shows an undershoot of 0.13 A. Moving to the period from $t = 15$ s to $t = 25$ s, the PISMCSTA and STA each show a current oscillation of 0.006 A and 0.012 A, respectively, and, from the period $t = 30$ s to $t = 45$ s (when the extreme loads are applied), a current oscillation equal to 0.015 A and 0.04396 A for each PISMCSTA and STA, respectively. According to these results, PISMCSTA is 66% higher than the STA.

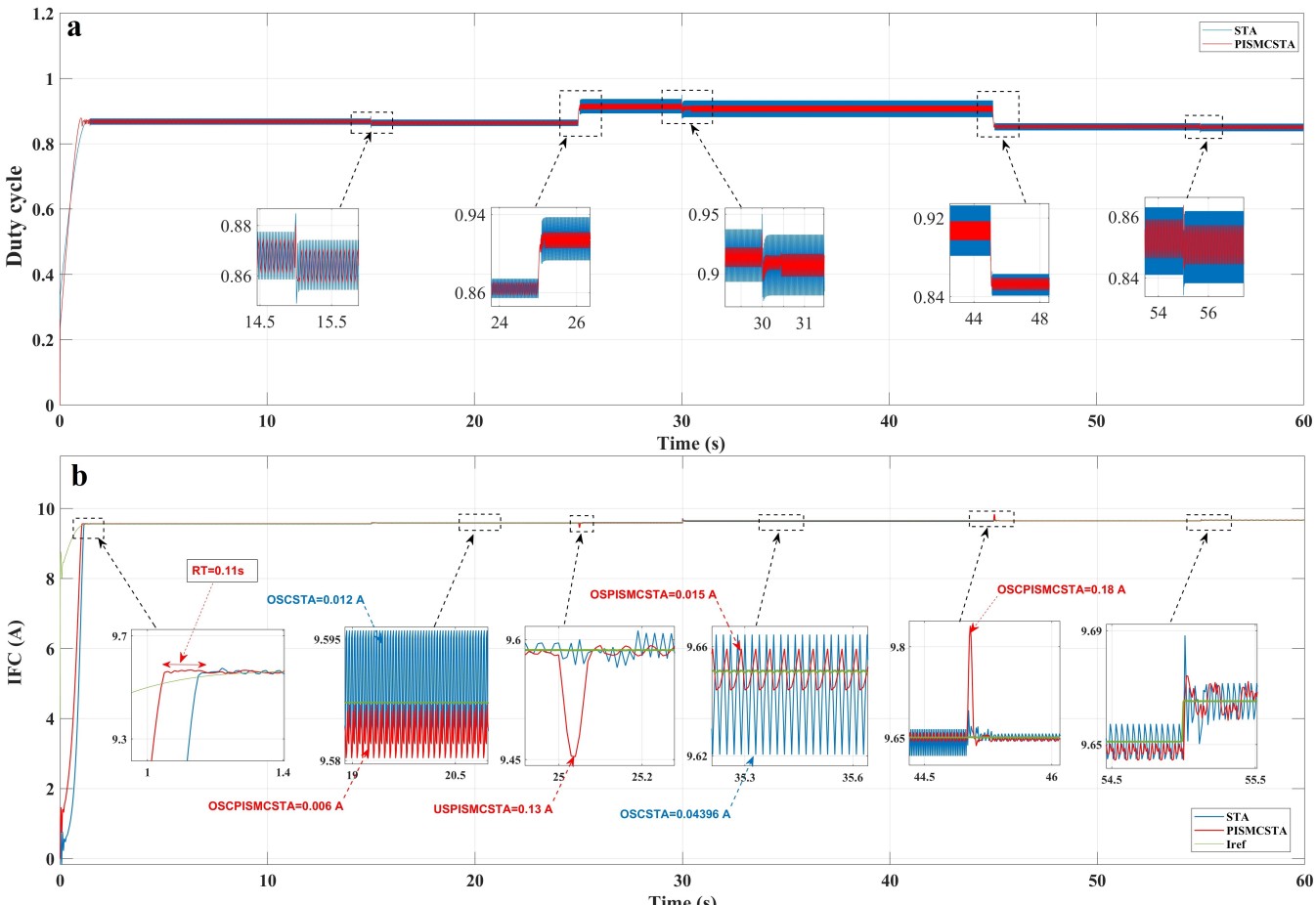

**Figure 11.** Controllers and PEMFC stack output signals: (**a**) duty cycle; (**b**) current.

The behavior of the output signals of the DC/DC boost converter under PISMCSTA and STA can be seen in Figure 12. The graphs illustrate that both controllers are able to achieve smooth and gradual movements toward the desired value. Additionally, it is evident from the figure that both controllers exhibit fast convergence. The results for current, voltage, and power show that there is minimal difference between the two controllers in terms of overshoot and undershoot. However, it is clearly seen that the PISMCSTA can effectively reduce the chattering effect by up to 68%, which is a significant improvement over STA.

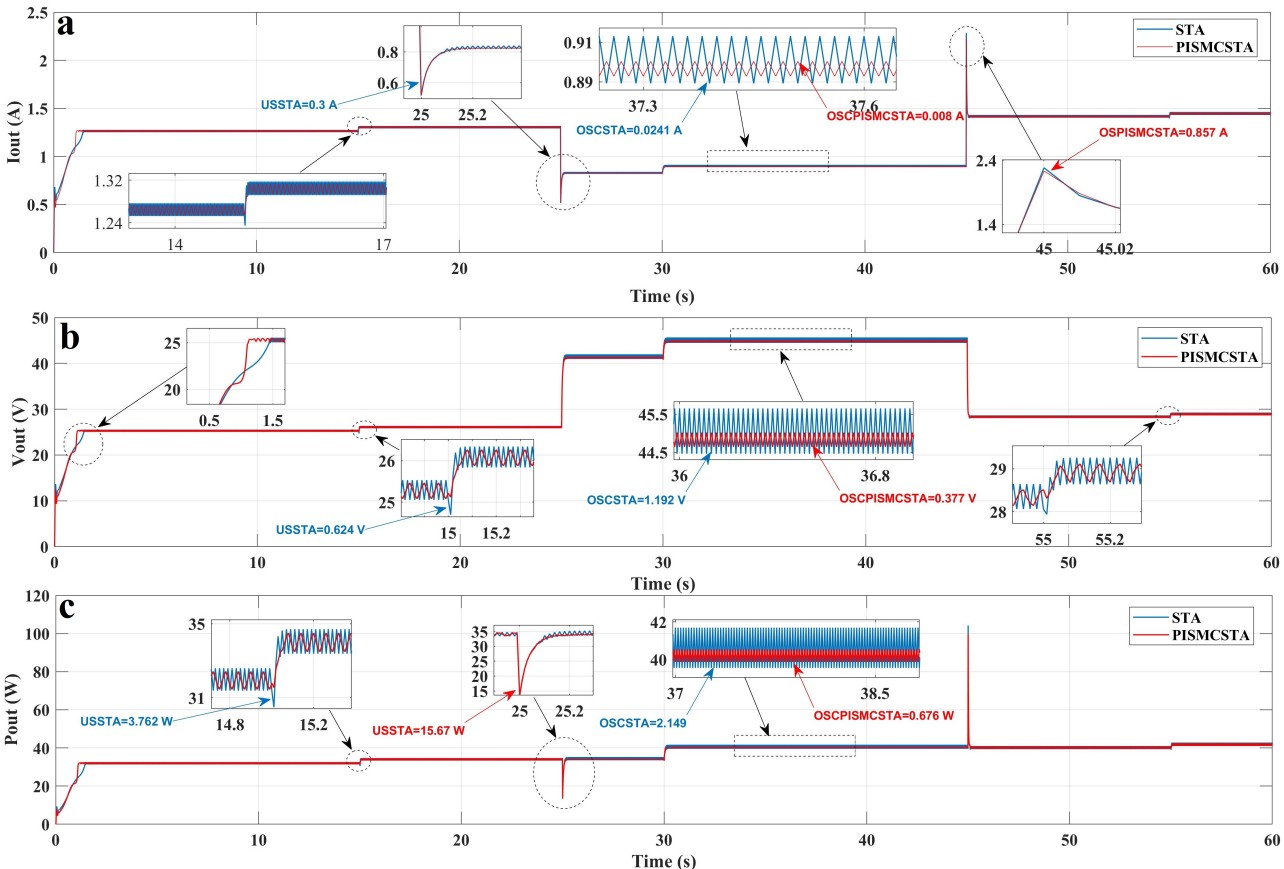

**Figure 12.** Boost converter outputs: (**a**) current; (**b**) voltage; (**c**) power.

## 7. Conclusions

In this paper, an innovative approach—PISMCSTA enhanced by BWOA—was designed to address the limitations of conventional control techniques in a PEMFC system. The effectiveness of this approach was subsequently compared with the established STA method in various operational conditions. Numerical simulation tests based on a PEMFC linked to a DC/DC boost converter were used to evaluate the performance of each technique.

Dynamic variations of load, temperature, and hydrogen gas pressure were induced at specific times. During the interval 25 s to 45 s, load changes were initiated, a shift in resistance from 20 Ω to 50 Ω occurred, and then subsequently regressed to 20 Ω during the last step. Temperature and hydrogen gas pressure variations were induced during the interval 15 s to 55 s: a shift in temperature from 30 °C to 40 °C, and a shift from 0.2 bar to 1 bar for the hydrogen gas pressure.

Simulation results showed that, during the dynamic variations, the PISMCSTA controller had superior performance in terms of accuracy, robustness, and chattering reduction. Compared to the other control strategy, PISMCSTA achieved a faster response time and lower chattering, by up to 66%. Thus, PISMCSTA can help mitigate unnecessary energy losses and improve the overall efficiency and stability of the system. The PISMCSTA controller is a promising solution for improving the performance and stability of the PEMFC system, which are important factors in many applications such as transportation, the energy sector, HVAC systems, smart grids, and autonomous systems.

**Author Contributions:** Conceptualization, M.Y.S.; methodology, M.Y.S.; software, M.Y.S.; validation, M.Y.S., O.B., J.A.C., P.A. and A.B.; formal analysis, M.Y.S. and A.B.; investigation, M.Y.S., O.B. and A.B.; resources, O.B.; data curation, M.Y.S., J.A.C., P.A. and A.B.; writing—original draft preparation, M.Y.S.; writing—review and editing, M.Y.S., O.B. and A.B.; visualization, M.Y.S., O.B., J.A.C., P.A. and A.B.; supervision, O.B.; project administration, O.B.; funding acquisition, O.B. All authors have read and agreed to the published version of the manuscript.

**Funding:** The authors wish to express their gratitude to the Basque Government through the project EKOHEGAZ II (ELKARTEK KK-2023/00051), to the Diputación Foral de Álava (DFA) through the project CONAVANTER, and to the UPV/EHU through the project GIU20/063 for supporting this work.

**Institutional Review Board Statement:** Not applicable.

**Informed Consent Statement:** Not applicable.

**Data Availability Statement:** Not applicable.

**Acknowledgments:** The authors wish to express their gratitude to the Basque Government, through the project EKOHEGAZ II (ELKARTEK KK-2023/00051), to the Diputación Foral de Álava (DFA), through the project CONAVANTER, and to the UPV/EHU, through the project GIU20/063, for supporting this work. Furthermore, the authors would like to acknowledge the projects supported by Telecommunications Signals and Systems Laboratory (TSS), University Amar Telidji, which played a vital role in making this research possible. Additionally, sincere appreciation is extended to all the associates who have directly or indirectly contributed to this work.

**Conflicts of Interest:** The authors declare no conflict of interest.

## Abbreviations

The following abbreviations are used in this manuscript:

| | |
|---|---|
| PI | Proportional–integral |
| SMC | Sliding mode control |
| PISMCSTA | PI sliding mode controller-based super-twisting algorithm |
| PEMFCs | Proton exchange membrane fuel cells |
| SGDM | Stochastic gradient descent with momentum |
| HOSM-TA | High-order sliding mode-based twisting algorithm |
| PID | Proportional–integral–derivative |
| FLC | Fuzzy logic controller |
| IFLC | Incremental fuzzy logic controller |
| AFLC | Adaptive fuzzy controller |
| NN | Neural network |
| NNFF | Neural network feed-forward |
| BP-NN | Back propagation neural network |
| ISE | Integral square error |
| IAE | Integral absolute error |
| ITAE | Integral time-weighted absolute error |
| PWM | Pulse-width modulation |
| BWOA | Black widow optimization algorithm |
| CR | Cannibalism rate |
| MR | Mutation rate |
| PR | Procreating rate |
| MPPT | Maximum power point-tracking |
| *P&O* | Perturb and observe |

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
