# Peer review of "PEMFC Current Control Using a Novel Compound Controller Enhanced by the Black Widow Algorithm: A Comprehensive Simulation Study"

_sustainability, doi:10.3390/su151813823_

Round 1

Reviewer 1 Report

The manuscript presents a novel control strategy called PISMCSTA for optimizing the net output power of PEMFC systems by reducing the chattering effect. The authors proposes a novel control strategy combining PI sliding mode control and super-twisting algorithm to optimize the net output power of PEM fuel cell systems by reducing the chattering effect and enhancing system performance and stability. The manuscript can be recommended for publication in Sustainability, after the authors addressing the following minor questions.

1. The introduction could be strengthened by providing more technical details on limitations of existing PEMFC control methods that motivate the need for the proposed strategy.

2. Additional comparison with other state-of-the-art optimization algorithms could help highlight benefits of using BWOA specifically.

3. The abstract concludes by briefly highlighting the key benefits and significance of the proposed PISMCSTA strategy. I suggest the authors add a statement in the end of abstract to point out the research significance.

4. In Figure 9, the subplots showing different polarization curves should be labeled as Fig. 9a, 9b, 9c, etc. The captions and discussion in the text should reference the specific subplot letters to make the analysis of the simulation results clearer.

The overall English level of this paper is good, but there are still a few grammatical errors. Minor editing of English language is required.

Author Response

Response to the reviewer comments

First of all, the author would express their sincere gratitude to the Editors and the Reviewers who gave us many constructive comments and valuable suggestions in order to improve this paper. The authors have revised the paper according to the reviewers’ comments and the changes made in the paper have been written in blue color. While the mistakes have been depicted in red color. The responses to the reviewer comments can be found below their respective comments.

Reviewer 1

Comments and Suggestions for Authors

The manuscript presents a novel control strategy called PISMCSTA for optimizing the net output power of PEMFC systems by reducing the chattering effect. The authors proposes a novel control strategy combining PI sliding mode control and super-twisting algorithm to optimize the net output power of PEM fuel cell systems by reducing the chattering effect and enhancing system performance and stability. The manuscript can be recommended for publication in Sustainability, after the authors addressing the following minor questions.

Point 1:  The introduction could be strengthened by providing more technical details on limitations of existing PEMFC control methods that motivate the need for the proposed strategy.

Response: Thank you for your comment. The authors have enhanced the introduction according to the review comments. Hence, additional paragraph is added to the revised manuscript.

“Despite the significant progress achieved in the field of proton exchange membrane fuel cell (PEMFC) control, several limitations such as the presence of uncertainties, control accuracy, robustness, and nonlinearities. However, many control methods have emerged that underscore the need for more advanced and robust strategies.” (State of the Art section)

(Contributions section) “The original contribution of this study lies in the development of a novel PI sliding mode controller-based super-twisting algorithm (PISMCSTA), which is designed to address the limitations of conventional control approaches in PEMFC systems. While existing methods have valuable attributes, such as simplicity, robustness, and adaptability, they often struggle to achieve an optimal balance between control accuracy, robustness, and the suppression of unwanted chattering effects in order to optimize the net output power of the PEMFC. This paper identifies a research gap that lies in the need for a comprehensive control solution that can simultaneously improve the accuracy, robustness, and stability of PEMFC systems under varying operating conditions.”

Point 2: Additional comparison with other state-of-the-art optimization algorithms could help highlight benefits of using BWOA specifically.

Response: Thank you for your comment. The authors have enhanced the introduction according to the review comments. Hence, additional paragraph is added to the revised manuscript. “As the quest for precise PEMFC models persists, evolutionary optimization methods have emerged as indispensable tools for robust parameter estimation. The innovative black widow optimization algorithm (BWOA) is introduced in response to this need. The algorithm's effectiveness is first validated through intricate benchmarking, showcasing its versatility in complex problem-solving scenarios. Subsequently, the BWOA algorithm is applied to extract critical parameters from PEMFC models across diverse operating temperatures. Comparative analysis ensues, contrasting parameter optimization outcomes achieved through BWO against five alternative state-of-the-art algorithms: particle swarm optimization (PSO), multi-verse optimizer (MVO), sine cosine algorithm (SCA), whale optimization algorithm (WOA) and grey wolf optimization (GWO). Through meticulous error analysis across two distinct PEMFC datasets, the research affirms the superior performance of the developed BWO algorithm compared to the alternative optimization approaches. Additionally, non-parametric testing confirms the consistent supremacy of BWO over the array of competing algorithms. To summarize, the study encompasses a comprehensive exploration of parameter optimization in PEMFCs, highlighting the significance of advanced optimization techniques. The novel BWOA, having showcased its efficacy in benchmarking and PEMFC parameter optimization, stands poised to enhance fuel cell control and efficiency while exemplifying algorithmic robustness within the field.”

Point 3:  The abstract concludes by briefly highlighting the key benefits and significance of the proposed PISMCSTA strategy. I suggest the authors add a statement in the end of abstract to point out the research significance.

Response: Thank you for your comment. The abstract has been revised and updated according to the reviewer comment.

Point 4:  In Figure 9, the subplots showing different polarization curves should be labeled as Fig. 9a, 9b, 9c, etc. The captions and discussion in the text should reference the specific subplot letters to make the analysis of the simulation results clearer.

Response: The figure and subplot are corrected according to the reviewer comments.

Point 5:  Comments on the Quality of English Language

The overall English level of this paper is good, but there are still a few grammatical errors. Minor editing of English language is required.

Response: The paper has been revised by a fluent speaker and many mistakes along the paper have been corrected. All changes are highlighted with blue-color in Revised Manuscript-Marked. For example:

Reviewer 2 Report

The manuscript is entitled "PEMFC Current Control Using Compound Sliding Mode Based Super-twisting and PI Controller Enhanced by Black Widow Algorithm". I do see good importance in this work; the topic is relevant. The work is well-written and well-presented. Important and useful discussions are presented as well. However, before I recommend its acceptance, some points must be clarified, and a minor revision is needed.

          1. The abstract is too general. The general pattern must be [problem definition] -> [concept] -> [action] -> [results] -> [significance], this is a good structure for enticing the readers and getting citations.

  1. Please revise the introduction to emphasize the work's original contribution, the scientific issue the study addresses, and the research gap/problem found.
  2. I encourage you to state the potential application of the new work. The reader gets a better impression of the future application of this technique.
  3. It is not clear what the contribution of the manuscript is to the empirical literature.
  4. There is no convincing link between the motivations for doing the paper and the way it has been conducted, as well as the conclusions reached.
  5. Please check spelling mistakes and the English language throughout the text.
  6. Limitations and challenges in the suggested approaches should be discussed in the conclusions section. The conclusion section appears to be just a detailed summary of the results and observations. All conclusions must be convincing statements on what was found to be impactful based on the solid support of the data, results, or discussion.

 Minor editing of the English language required

Author Response

Response to the reviewer comments

First of all, the author would express their sincere gratitude to the Editors and the Reviewers who gave us many constructive comments and valuable suggestions in order to improve this paper. The authors have revised the paper according to the reviewers’ comments and the changes made in the paper have been written in blue color. While the mistakes have been depicted in red color. The responses to the reviewer comments can be found below their respective comments.

Reviewer 2

Comments and Suggestions for Authors

The manuscript is entitled "PEMFC Current Control Using Compound Sliding Mode Based Super-twisting and PI Controller Enhanced by Black Widow Algorithm". I do see good importance in this work; the topic is relevant. The work is well-written and well-presented. Important and useful discussions are presented as well. However, before I recommend its acceptance, some points must be clarified, and a minor revision is needed.

Point 1:  The abstract is too general. The general pattern must be [problem definition] -> [concept] -> [action] -> [results] -> [significance], this is a good structure for enticing the readers and getting citations.

Response: Thank you for your comment. The authors have enhanced the abstract according to the reviewer comments.

Point 2:  Please revise the introduction to emphasize the work's original contribution, the scientific issue the study addresses, and the research gap/problem found.

Response: Thank you for your comment.  The authors have revised the introduction according to the reviewer comments. An additional paragraph is added to the revised manuscript. “The original contribution of this study lies in the development of a novel PI sliding mode controller-based super-twisting algorithm (PISMCSTA) designed to address the limitations of conventional control approaches in PEMFC systems. While existing methods exhibit valuable attributes, such as simplicity, robustness, and adaptability, they often struggle to strike an optimal balance between control accuracy, robustness, and the suppression of undesired chattering effects. This paper identifies a research gap that lies in the need for a comprehensive control solution that can simultaneously enhance the accuracy, robustness, and stability of PEMFC systems under varying operational conditions.”

Point 3:  I encourage you to state the potential application of the new work. The reader gets a better impression of the future application of this technique.

Response: Thank you for your comment.  The authors have added a potential application of the new work to the revised manuscript (conclusion). An additional paragraph is added to the revised manuscript.  “In this paper, an innovative approach PISMCSTA enhanced by BWOA was designed to address the limitations of conventional control techniques in PEMFC system. The effectiveness of this approach was subsequently compared with the established STA method in various operational conditions. Numerical simulation tests based on a PEMFC linked to a DC/DC boost converter were used to evaluate the performance of each technique.

Dynamic variations of load, temperatures and hydrogen gas pressures were induced at specific times. At interval of 25 s to 45 s, load changes were initiated, a shift in resistance from 20 Ω to 50 Ω and subsequently regressed to 20 Ω during the last step. Temperature and hydrogen gas pressure variations were induced during the interval 15 s to 55 s, a shift in temperatures from 30 to 40 degrees and a shift from 0.2 bar to 1 bar for the hydrogen gas pressure.

Simulation results showed that during the dynamic variations, the PISMCSTA controller had superior performance in terms of accuracy, robustness and chattering reduction. Compared to the other control strategy, PISMCSTA achieves a faster response time and lower chattering up to 66%. This latter can help mitigate unnecessary energy losses and improve the overall efficiency and stability of the system. The PISMCSTA controller is a promising solution for improving the performance and stability of the PEMFC system, which are important for many applications such as the transportation, energy sectors, HVAC systems, smart grids and autonomous systems.”

Point 4:  It is not clear what the contribution of the manuscript is to the empirical literature.

Response: Thank you for your comment.  The contribution of the paper has been clearly presented in the introduction of the revised manuscript (contribution subsection). We think that the paper has an important contribution since it proposed a novel algorithm aiming to improve the performance of the fuel cell system. A comparison study of the STA (high order sliding mode) has been performed to validate the proposed PISMCSTA controller. Moreover, it should be noted that this theoretical study has been done based on the technical data of a commercial Heliocenris fuel cell system since a practical validation of the proposed algorithms will be done later. On the other hand, we think that the quality of the revised manuscript is significantly improved since we have carefully revised the manuscript based on the reviewer’s suggestions. An additional paragraph is added to the revised manuscript. “The original contribution of this study lies in the development of a novel PI sliding mode controller-based super-twisting algorithm (PISMCSTA) designed to address the limitations of conventional control approaches in PEMFC systems. While existing methods exhibit valuable attributes, such as simplicity, robustness, and adaptability, they often struggle to strike an optimal balance between control accuracy, robustness, and the suppression of undesired chattering effects therefore optimizing the PEMFC net output power. This paper identifies a research gap that lies in the need for a comprehensive control solution that can simultaneously enhance the accuracy, robustness, and stability of PEMFC systems under varying operational conditions.”

Point 5:  There is no convincing link between the motivations for doing the paper and the way it has been conducted, as well as the conclusions reached.

Response: Thank you for your comment.  The authors have enhanced the introduction and the conclusion according to the reviewer’s comment. However, the primary motivation behind this research paper was to address the limitations and challenges faced by existing control strategies in the context of proton exchange membrane fuel cell (PEMFC) systems. Previous studies have highlighted issues such as the chattering effect and suboptimal power operation. The introduction of the PI sliding mode controller-based super-twisting algorithm (PISMCSTA) aimed to bridge this gap by introducing a novel control strategy that combines the advantages of sliding mode control, PI control, and super-twisting algorithm. This approach was intended to simultaneously optimize power output while reducing the chattering effect that can lead to instability. Throughout the paper, we systematically detailed the development and implementation of the PISMCSTA, providing technical insights into its design, structure, and application. The concept behind the PISMCSTA was founded on the integration of proven control techniques, coupled with the novel inclusion of the black widow optimization algorithm (BWOA) to enhance controller tuning in response to disturbances. Theoretical foundations were laid out for the proposed controller's operation, highlighting its key features and advantages. Numerical simulations were carried out to empirically demonstrate the efficacy of the PISMCSTA. Comparative results were presented, comparing its performance against conventional strategies. Notably, the PISMCSTA showcased remarkable reductions in response time and chattering effect, as well as superior tracking accuracy and stability. These findings validated the viability of the proposed approach and reinforced its potential to outperform conventional methods. Ultimately, the conclusions reached in this research paper closely align with the initial motivations. The PISMCSTA directly addresses the need for improved performance, stability, and efficiency in PEMFC systems. By skillfully amalgamating established control methodologies and introducing innovative tuning techniques, the PISMCSTA offers a holistic solution to the limitations observed in previous control strategies. Its ability to substantially mitigate the chattering effect while optimizing power output signifies a significant stride towards realizing more reliable and effective PEMFC systems.

Point 6:  Please check spelling mistakes and the English language throughout the text.

Response: The paper has been revised by a fluent speaker and many mistakes along the paper have been corrected. All changes are highlighted with blue-color in Revised Manuscript-Marked.

Point 7:  Limitations and challenges in the suggested approaches should be discussed in the conclusions section. The conclusion section appears to be just a detailed summary of the results and observations. All conclusions must be convincing statements on what was found to be impactful based on the solid support of the data, results, or discussion.

Response: Thank you for your comment.  The authors have the conclusion according to the reviewer’s comment.

Comments on the Quality of English Language

 Minor editing of the English language required

Response: The paper has been revised by a fluent speaker and many mistakes along the paper have been corrected. All changes are highlighted with blue-color in Revised Manuscript-Marked. For example:

Reviewer 3 Report

Dear Authors,

General Comment

1.

The main theme of this paper is ecology (reduction of the thermal effect). The disadvantage of PEM fuel cells is that they generate significant amounts of heat during operation. It's a contradiction.

2.

The second significant disadvantage - it's only a simulation. A computer simulation is just a quasi-mathematical representation of the real world. Only real design can verify these assumptions. There are losses in the process, after all.

3.

In my opinion, the proposed method [46] is controversial.

My Conclusion

In my opinion, improving the work of the PEM fuel cell is uncertain. The Authors should include the economic aspect. They should also prove their thesis through a practical application. It's a simulation, not an actual measurement. I really hope you can prove that.

-

Author Response

Response to the reviewer comments

First of all, the author would express their sincere gratitude to the Editors and the Reviewers who gave us many constructive comments and valuable suggestions in order to improve this paper. The authors have revised the paper according to the reviewers’ comments and the changes made in the paper have been written in blue color. While the mistakes have been depicted in red color. The responses to the reviewer comments can be found below their respective comments.

Reviewer 3

Comments and Suggestions for Authors

Dear Authors,

General Comment

Point 1: The main theme of this paper is ecology (reduction of the thermal effect). The disadvantage of PEM fuel cells is that they generate significant amounts of heat during operation. It's a contradiction.

Response: Thank you for your comment. While it is true that PEM fuel cells generate heat during operation, our research focuses on addressing the ecological aspect by improving the overall efficiency and performance of these fuel cells. The primary goal of our study is to reduce the thermal effects that can negatively impact the environment and system performance. The generation of heat in PEMFCs is indeed a known issue, and it can lead to various challenges, including decreased efficiency and the need for additional cooling systems, which can be energy-intensive. Our proposed PISMCSTA aims to mitigate these challenges by optimizing the operation of the fuel cell system and reducing the chattering effect. This finally, has detrimental effects on electrical circuits, leading to increased heat generation and energy losses within the system. Moreover, chattering can contribute to system instability, which, in turn, can result in suboptimal energy conversion and potential environmental repercussions. By mitigating the chattering effect through our proposed PISMCSTA, we are not only enhancing the performance and efficiency of PEMFCs but also indirectly contributing to a more ecologically responsible approach. This is because a smoother, more controlled operation reduces the stress on electrical components, lowers energy losses, and minimizes the risk of system instability, ultimately leading to a more sustainable and eco-friendly utilization of PEMFC technology. This, in turn, helps in reducing the thermal effect because a more efficiently controlled system can operate with reduced heat generation. In essence, our research addresses the ecological concerns associated with PEM fuel cells by improving their efficiency and control, ultimately contributing to a more sustainable and environmentally friendly energy solution. We hope this clarification helps to reconcile the apparent contradiction and underscores the ecological significance of our work.

Point 2: The second significant disadvantage - it's only a simulation. A computer simulation is just a quasi-mathematical representation of the real world. Only real design can verify these assumptions. There are losses in the process, after all.

Response: Thank you for your comment. The authors fully recognize the significance of experimental validation as a critical step in verifying the assumptions made in any research, including ours. While computer simulations indeed provide a quasi-mathematical representation of the real world and may not fully encompass all real-world complexities, they play a vital role in informing the design and feasibility assessment of new strategies before transitioning to practical implementation. In the context of our research, utilizing computer simulations serves several purposes. Firstly, simulations provide an environment where we can systematically study the proposed PISMCSTA under controlled conditions. This allows us to evaluate the algorithms' performance, analyze their effects on key metrics, and optimize their parameters efficiently. Additionally, simulations enable us to conduct a wide range of scenario-based analyses, which may be impractical or cost-prohibitive to replicate experimentally. They also facilitate the assessment of the proposed approach's sensitivity to various operating conditions and disturbances, giving us valuable insights into its behavior across different scenarios. Also, It should be noted that this theoretical study has been done based on the technical data of a commercial Heliocenris fuel cell system, as practical validation of the proposed algorithms will be conducted at a later stage. We absolutely agree that experimental validation is essential, and we do plan to progress toward real-world verification to validate the findings of our simulation-based research. Our intention is to use the simulation results as a guide to inform the design of practical experiments. By iteratively refining and testing our strategies through both simulations and experiments, we aim to bridge the gap between theoretical concepts and real-world applications, ultimately contributing to more robust and reliable solutions.

Point 3: In my opinion, the proposed method [46] is controversial. Ref

Response: Thank you for the comment. The section 5 (optimization using black widow algorithm), has been revised and enhanced according to the reviewer’s comment.

The black widow optimization algorithm (BWOA), like other metaheuristic algorithms, can be applied to a wide range of optimization problems, including: engineering optimization, machine learning, function optimization, robotics and the parameter tuning such in our case. In the context of parameter tuning the BWOA can be used to optimize the control parameters of renewable energy systems [2,3,4], such as the settings for solar tracking systems or the control strategy for PEMFCs and wind turbines…. etc. This can help maximize energy capture and system efficiency [5].

The proposed BWOA is used in order to tune the PISMCSTA and STA controller’s parameters. Iref  is the reference current extracted by the P&O technique. IL is the current produced by the PEMFC power system. e is error current, denotes the difference between IL and Iref . During each iteration of the BOWA, a population values of Kp,i,1,2,3 for the PISMCSTA and  K1,2 for the STA are generated, which are then substituted into the objective function ITAE. PISMCSTA and STA controllers take e as the input signal and then produce the corresponding control signal. The process is repeated until the error signal approaches zero. Finally, the best values of the parameters will be identified and used to design the optimum PISMCSTA and STA controllers. 

Please see references of:

[1]: Hayyolalam, V., & Kazem, A. A. P. (2020). Black widow optimization algorithm: a novel meta-heuristic approach for solving engineering optimization problems. Engineering Applications of Artificial Intelligence, 87, 103249.

[2]: Munagala, V. K., & Jatoth, R. K. (2021). Optimal Design of Fractional Order PID Controller for AVR System Using Black Widow Optimization (BWO) Algorithm. In Machine Learning, Deep Learning and Computational Intelligence for Wireless Communication: Proceedings of MDCWC 2020 (pp. 19-34). Springer Singapore.

[3]: Munagala, V. K., & Jatoth, R. K. (2022). Improved fractional PIλDμ controller for AVR system using Chaotic Black Widow algorithm. Computers & Electrical Engineering, 97, 107600.

[4]: Mathur, N., Meena, V. P., & Singh, V. P. (2022). Black widow optimisation-based controller design for Riverol-Pilipovik water treatment system. International Journal of Modelling, Identification and Control, 40(3), 204-209.

[5]: Dahiya, P., & Saha, A. K. (2022). Frequency regulation of interconnected power system using black widow optimization. IEEE Access, 10, 25219-25236.

My Conclusion

In my opinion, improving the work of the PEM fuel cell is uncertain. The Authors should include the economic aspect. They should also prove their thesis through a practical application. It's a simulation, not an actual measurement. I really hope you can prove that.

Reviewer 4 Report

Should be improved.

Author Response

Response to the reviewer comments

First of all, the author would express their sincere gratitude to the Editors and the Reviewers who gave us many constructive comments and valuable suggestions in order to improve this paper. The authors have revised the paper according to the reviewers’ comments and the changes made in the paper have been written in blue color. While the mistakes have been depicted in red color. The responses to the reviewer comments can be found below their respective comments.

Reviewer 4

The content of this article is quite interesting. However, the authors should correct some of the following issues.

Point 1: The word “In contrast” is inappropriate in the abstract. It should be changed to “Additionally” or “In addition.”

Response: Thank you for your comment. The abstract has been revised and updated according to the reviewer comment.

Point 2: The abstract is relatively short. The authors should emphasize the content made in the article. In addition, they should point out how the PISMCSTA algorithm differs from existing ones.

Response: Thank you for your comment. The abstract has been enhanced, revised and updated according to the reviewer comment.

Point 3: Should not use keywords that are too long and complicated.

Response: Thank you for your comment. The key words are updated

Point 4: The combination of PI and SMC algorithms can bring high efficiency. This combination has been applied to several mechatronic systems. The authors should give some introduction related to them; see: A novel approach with a fuzzy sliding mode proportional integral control algorithm tuned by fuzzy method (FSMPIF); Proposing an original control algorithm for the active suspension system to improve vehicle vibration: Adaptive fuzzy sliding mode proportional integral-derivative tuned by the fuzzy (AFSPIDF).

Response: Thank you for your comment. The authors have enhanced the introduction according to the review comments.

Point 5: In the Introduction section, the authors should clarify the novelty and difference of the algorithm designed in this article.

Response: Thank you for your comment. The authors have taken in considerations the reviewer comments about the novelty and difference of the algorithm. Hence, an additional paragraph is added to the revised manuscript (Contribution subsection).  “The original contribution of this study lies in the development of a novel PI sliding mode controller-based super-twisting algorithm (PISMCSTA), which is designed to address the limitations of conventional control approaches in PEMFC systems. While existing methods have valuable attributes, such as simplicity, robustness, and adaptability, they often struggle to achieve an optimal balance between control accuracy, robustness, and the suppression of unwanted chattering effects in order to optimize the net output power of the PEMFC. This paper identifies a research gap that lies in the need for a comprehensive control solution that can simultaneously improve the accuracy, robustness, and stability of PEMFC systems under varying operating conditions.”

Point 6: The statement “This paper is divided into three sections” is incorrect. This article is divided into 7 sections, not 3 sections.

Response: Thank you for your comment. The authors have corrected the statements according to the reviewer comment.

Point 7: In Table 3, units should be separated into a distinguishing column.

Response: Thank you for your comment. The authors have enhanced the Table 3 (Table 4 for the new version of manuscript) according to the review comments.

Point 8: Why do the authors only use the PI algorithm instead of the PID?

Response: Thank you for your comment. In this context, the authors choose to use the PI surface (instead of PID) in order to avoid the added of an additional derivative action, for the second error derivative, this can introduce the calculation complexities that may not be warranted for our specific research objectives and also the added of the derivative leads to a very noisy signal that which cannot be easily differentiated. This finally, can lead the system to be less stable because of the high sensitivity of the derivative term to the noisy signals.  In addition, the PI controller provides integral action that can effectively address steady-state errors, which can be crucial for ensuring stable and accurate control of the fuel cell systems output power. By using a PI controller, we aimed to strike a balance between control performance and complexity, especially when considering real-world implementation challenges.

Point 9: Conclusion should be written in 2 paragraphs instead of 1 single paragraph. The authors should point out the advantages and drawbacks of the algorithm established in this manuscript.

Response: Thank you for your comment. The conclusion has been revised and updated according to the reviewer comment.

Point 10: Comments on the Quality of English Language

Should be improved

Response: The paper has been revised by a fluent speaker and many mistakes along the paper have been corrected. All changes are highlighted with blue-color in Revised Manuscript-Marked. For example

Reviewer 5 Report

The submitted manuscript presents the design of a PI sliding mode control super twisting algorithm applied to a boost converter connected to Proton Exchange Membrane Fuel Cells. The researched topic is of interest for the communities of renewable energies and power electronics. However, some comments are due.

- English should be deeply revised. Many typos, errors, or wrong usage of verbs can be found. The Reviewer suggests deeply revising the manuscript before resubmitting.

- Authors should enhance the Introduction by clearly mentioning the advances of their approach with respect to the existing state of the art. Moreover, despite dedicating a subsection to the state of the art, the Reviewer believes it could be further improved by mentioning more recent works related to supertwisting, or more in general, higher-order sliding mode control algorithms applied to the control of DC/DC converters. Some suggestions are [A Saturated Higher Order Sliding Mode Control Approach for DC/DC Converters, 2022 IEEE 17th International Conference on Control & Automation (ICCA), 2022, pp. 44-49] where a novel saturation algorithm for higher order SMC algorithms is presented and [Generalized Super-Twisting control of a Dual Active Bridge for More Electric Aircraft, 2021 European Control Conference (ECC), 2021, pp. 1610-1615] where generalized super-twisting control is applied to a DAB onboard an electrified aircraft.

- Minor correction: the last paragraph of Section 1 does not cite Section 5

- In Section 4.2, the first sentence is incorrect. The equivalent control preserves the system in the surface (in fact, the equivalent control is retrieved by supposing s=0 and \dot{s}=0) while the switching term is used to bring the system to the sliding surface

- Before equation (19) Authors should clearly state what is the value of the relative degree of the studied system. Is it 1 or 2 or 3 or ...?

- Equation (20) seems to be incorrect. The one presented in the manuscript is the formula of s_1, not \dot{s}_1. When you want to compute \dot{s}_1, \ddot{e} should appear thus yielding the presence of \dot{u} in the right hand side. Authors should definitely fix this equation as it is fundamental for subsequent analysis. (note that, using the LaTeX style, \dot{s} indicates the time derivative of s, while \ddot{e} indicates the double time derivative of e) 

- It is initially assumed that the reference current changes with time according to the MPPT algorithm. However, looking at (20), it is clear that the Authors considered the current reference to be constant since it does not appear when computing \dot{s}_1. Please, explain your reasoning.

- The computation of \dot{V}_1 in (32) is invalid due to the wrong computetion of \dot{s}_1. Authors need to correctly write \dot{s}_1 and perform again the computation. In any case, remove the dot in the integrals before dt.

- Before concluding section 5, Authors need to better explain how the BWOA is applied to tuning of PISMCSTA parameters and report the ITAE function. Which parameters of the PISMCSTA were used in the BWOA?

-  From Figure 3 it seems that the BWOA algorithm updates the parameters online during simulation as the BWOA needs the error signal to compute the ITAE function and, in turns, the controller parameters. However, the Authors claim that the values are computed offline and they provide fixed values in Table 3 that are used in simulation. Please explain better this point.

- Regarding the results shown if Section 6, they seem to be obtained using Simulink. Which toolbox was used (e.g. SimPowerSystem, SimScape or simply implementing the system equations in Simulink)?

- Considering Figure 12, what is the switching frequency of the PWM? 

- It would be interesting to see the duty cycle for the STA and PISMCSTA algorithms

English should be deeply revised. Many typos, errors, or wrong usage of verbs can be found. The Reviewer suggests deeply revising the manuscript before resubmitting.

Author Response

Response to the reviewer comments

First of all, the author would express their sincere gratitude to the Editors and the Reviewers who gave us many constructive comments and valuable suggestions in order to improve this paper. The authors have revised the paper according to the reviewers’ comments and the changes made in the paper have been written in blue color. While the mistakes have been depicted in red color. The responses to the reviewer comments can be found below their respective comments.

Reviewer 5

The submitted manuscript presents the design of a PI sliding mode control super twisting algorithm applied to a boost converter connected to Proton Exchange Membrane Fuel Cells. The researched topic is of interest for the communities of renewable energies and power electronics. However, some comments are due.

Point 1:  English should be deeply revised. Many typos, errors, or wrong usage of verbs can be found. The Reviewer suggests deeply revising the manuscript before resubmitting.

Response: The paper has been revised by a fluent speaker and many mistakes along the paper have been corrected. All changes are highlighted with blue-color in the Revised Manuscript-Marked.

Point 2: Authors should enhance the Introduction by clearly mentioning the advances of their approach with respect to the existing state of the art. Moreover, despite dedicating a subsection to the state of the art, the Reviewer believes it could be further improved by mentioning more recent works related to supertwisting, or more in general, higher-order sliding mode control algorithms applied to the control of DC/DC converters. Some suggestions are [A Saturated Higher Order Sliding Mode Control Approach for DC/DC Converters, 2022 IEEE 17th International Conference on Control & Automation (ICCA), 2022, pp. 44-49] where a novel saturation algorithm for higher order SMC algorithms is presented and [Generalized Super-Twisting control of a Dual Active Bridge for More Electric Aircraft, 2021 European Control Conference (ECC), 2021, pp. 1610-1615] where generalized super-twisting control is applied to a DAB onboard an electrified aircraft.

Response: Thank you for your comment. The authors have enhanced the introduction according to the review comments.

Point 3: Minor correction: the last paragraph of Section 1 does not cite Section 5

Response: Thank you for your comment. The authors corrected the paragraph of Section 1 according to the reviewer comment.

Point 4:  In Section 4.2, the first sentence is incorrect. The equivalent control preserves the system in the surface (in fact, the equivalent control is retrieved by supposing s=0 and \dot{s}=0) while the switching term is used to bring the system to the sliding surface

Response: Thank you for your comment. The authors corrected the paragraph according to the reviewer comment.

Point 5:  Before equation (19) Authors should clearly state what is the value of the relative degree of the studied system. Is it 1 or 2 or 3 or ...?

Response: Thank you for your comment. The authors set r equal to 2 in order to simplicity. Hence, higher values of the relative degree r, a more differentiations are required in the control law which can lead to more complex control algorithms. Designing controllers with high relative degrees might require additional sensors or measurements, and the resulting control law could be more intricate in real word applications. Additional paragraph is added to the revised manuscript (before Eq 19).  “Therefore, by setting the relative degree r equal to 2, the sliding surface for the first proposed controller (STA) can be expressed by equation 19.”

Point 6: Equation (20) seems to be incorrect. The one presented in the manuscript is the formula of s_1, not \dot{s}_1. When you want to compute \dot{s}_1, \ddot{e} should appear thus yielding the presence of \dot{u} in the right hand side. Authors should definitely fix this equation as it is fundamental for subsequent analysis. (note that, using the LaTeX style, \dot{s} indicates the time derivative of s, while \ddot{e} indicates the double time derivative of e) 

Response: Thank you for your comment. Unfortunately, written mistakes have been made in eq.18 and 19 of the old manuscript versions. However, these mistakes are corrected in the revised manuscript.   

Point 7:  It is initially assumed that the reference current changes with time according to the MPPT algorithm. However, looking at (20), it is clear that the Authors considered the current reference to be constant since it does not appear when computing \dot{s}_1. Please, explain your reasoning.

Response: Thank you for the comment. In our control strategy, we initially considered the dynamics of the reference current (iref ) as it changes over time due to the P&O MPPT technique. However, upon closer examination of the system behavior and its impact on the control strategy, we have opted to simplify our model by omitting the derivative of the reference current from the equation for . The reason for this simplification lies in the relative magnitudes of the various terms involved in the control dynamics. After careful analysis, we have determined that the rate of change of the reference current is either relatively small or has minimal influence on the overall system dynamics. In essence, the reference current can be treated as a slowly varying or quasi-constant parameter in comparison to other more significant system variables, such as the inductor current (iL ) and control inputs. By removing the derivative of the reference current from the equation for  , we simplify the control model without sacrificing the accuracy of our control strategy. This simplification streamlines our calculations and provides a clearer representation of the dominant dynamics affecting the sliding surface derivative ​. An additional paragraph is added to the revised manuscript (after Eq 19).  

“In the context of the proposed control strategies, the dynamics of the reference current (iref ) change over the time due to the P\&O MPPT technique. However, after a closer examination of the system behavior and its impact on the control strategy, the decision was made to simplify without sacrificing the control strategies by omitting the derivative of the reference current. Additionally, the analysis determined that the rate of change of the iref either has minimal influence on the overall system dynamics or is relatively small. Essentially, when compared to more significant system variables such as inductor current (iL ) and the control inputs, the iref can be treated as a slowly varying or quasi-constant parameter.”

Point 8: The computation of \dot{V}_1 in (32) is invalid due to the wrong computetion of \dot{s}_1. Authors need to correctly write \dot{s}_1 and perform again the computation. In any case, remove the dot in the integrals before dt.

Response: Thank you for the comment. The equations have been revised and the dot has been removed.

Point 9: Before concluding section 5, Authors need to better explain how the BWOA is applied to tuning of PISMCSTA parameters and report the ITAE function. Which parameters of the PISMCSTA were used in the BWOA?

Response: Thank you for the comment. The authors have taken in considerations the reviewer comments about the BOWA technique. Hence, figure 6 is updated and an additional paragraph is added to the revised manuscript (Section 5).  In this simulation, the BWOA is used in order to tune the PISMCSTA and STA controller’s parameters. Iref  is the reference current extracted by the P&O technique. IL is the current produced by the PEMFC power system. e is error current, denotes the difference between IL  and Iref . During each iteration of the BOWA, a population values of Kp,i,1,2,3 for the PISMCSTA and  K1,2 for the STA are generated, which are then substituted into the ITAE objective function ITAE. PISMCSTA and STA controllers take e as the input signal and then produce the corresponding control signal. The process is repeated until the error signal approaches zero. Finally, the best values of the parameters will be identified and used to design the optimum PISMCSTA and STA controllers. 

Point 10:  From Figure 3 it seems that the BWOA algorithm updates the parameters online during simulation as the BWOA needs the error signal to compute the ITAE function and, in turns, the controller parameters. However, the Authors claim that the values are computed offline and they provide fixed values in Table 3 that are used in simulation. Please explain better this point.

Response: Thank you for the comment. The figure 3 has been updated to the new version of the manuscript.

Point 11:  Regarding the results shown if Section 6, they seem to be obtained using Simulink. Which toolbox was used (e.g. SimPowerSystem, SimScape or simply implementing the system equations in Simulink)?

Response: Thank you for the comment. The authors use the SimPowerSystem.

Point 12:  Considering Figure 12, what is the switching frequency of the PWM? 

Response: Thank you for the comment. A DC/DC boost converter parameter (Table 2) are added to the section 3 of the revised manuscript.

Point 13:  It would be interesting to see the duty cycle for the STA and PISMCSTA algorithms

Response: Thank you for the comment. The duty cycle signals for PISMCSTA and STA have been depicted in Figure 10a.

Comments on the Quality of English Language

English should be deeply revised. Many typos, errors, or wrong usage of verbs can be found. The Reviewer suggests deeply revising the manuscript before resubmitting.

Response: Thank you for the comment. The English has been revised by a fluent speaker according to the reviewer comments.

Round 2

Reviewer 3 Report

Dear Authors,

General Comment

Thank you very much for your response.

I'll repeat my first comment; this is just a simulation of measurement under ideal conditions. I suggest changing the title - "Simulation of the PEMFC Current Control Using..."

The revised version of the paper is already satisfactory, but the experimental part is still missing. I would suggest you consider publishing the next part of this paper in the future.

My Conclusion

This material can be a contribution to further discussion in a wider group. All assumptions will be verified by a practical application.

Author Response

Response to the reviewer comments

First of all, the author would express their sincere gratitude to the Editors and the Reviewers who gave us many constructive comments and valuable suggestions in order to improve this paper. The authors have revised the paper according to the reviewers’ comments and the changes made in the paper have been written in blue color. While the mistakes have been depicted in red color. The responses to the reviewer comments can be found below their respective comments.

Reviewer 3

General Comment
Thank you very much for your response.

Point 1: I'll repeat my first comment; this is just a simulation of measurement under ideal conditions. I suggest changing the title - "Simulation of the PEMFC Current Control Using..."

Response: Thank you for your comment. The title has been changed and updated (revised manuscript) according to the reviewer comment.

The revised version of the paper is already satisfactory, but the experimental part is still missing. I would suggest you consider publishing the next part of this paper in the future.

My Conclusion
This material can be a contribution to further discussion in a wider group. All assumptions will be verified by a practical application.

Reviewer 4 Report

Now, this article can be accepted!

Author Response

Response to the reviewer comments

First of all, the author would express their sincere gratitude to the Editors and the Reviewers who gave us many constructive comments and valuable suggestions in order to improve this paper. The authors have revised the paper according to the reviewers’ comments and the changes made in the paper have been written in blue color. While the mistakes have been depicted in red color. The responses to the reviewer comments can be found below their respective comments.

Reviewer 4

Point: Now, this article can be accepted!

Response: Thank you for your valuable comments

Reviewer 5 Report

Authors have correctly replied the Reviewer's questions. No further changes are required.

Only minor fixes should be applied to the quality of English language.

Author Response

Response to the reviewer comments

First of all, the author would express their sincere gratitude to the Editors and the Reviewers who gave us many constructive comments and valuable suggestions in order to improve this paper. The authors have revised the paper according to the reviewers’ comments and the changes made in the paper have been written in blue color. While the mistakes have been depicted in red color. The responses to the reviewer comments can be found below their respective comments.

Reviewer 5

Comments and Suggestions for Authors

Point 1: Authors have correctly replied the Reviewer's questions. No further changes are required.

Response: Thank you for your valuable comments.

Comments on the Quality of English Language

Point 2: Only minor fixes should be applied to the quality of English language

Response: Thank you for your comment. The English has been revised.  For example, in:

Section 2, line 238, the statement “In the presence of platinum, also a catalytic reduction occurs, occurs, the oxygen combined with the protons which that have crossed the electrolyte membrane and the electrons arriving from the external circuit.” are revised in the same line of the revised manuscript as follows: “In the presence of platinum, a catalytic reduction occurs, occurs, the oxygen combined with the protons that have crossed the electrolyte membrane and the electrons arriving from the external circuit.”.

Line 252, the statement “The activation polarization occurs due to the electrochemical reactions which required a certain amount of energy to overcome the energy barrier for the electrochemical reaction to proceed [34]:.” are revised in the same line of the revised manuscript as follows: “The activation polarization occurs due to the electrochemical reactions which require a certain amount of energy to overcome the energy barrier for the electrochemical reaction to proceed [34]:

Line 259, the statement “The ohmic polarization results from the electron transfer resistance across the collector plates and carbon electrodes, denote RC and the proton movement resistance across the solid membrane which denoted RM. The equivalent membrane resistance is given as follows [34]:.” are revised in the same line of the revised manuscript as follows: “statement “The ohmic polarization results from the electron transfer resistance across the collector plates and carbon electrodes, denoted RC and the proton movement resistance across the solid membrane which denoted RM. The equivalent membrane resistance is given as follows [34].”

Section 3, line 282, the statement “In many applications related to clean energy, fuel cells are combined with power converters which yield an efficient path from the cell stack to the load and it will also deliver a regulated output voltage [35].” are revised in the same line of the revised manuscript as follows: “In many applications related to clean energy, fuel cells are combined with power converters which produce an efficient path from the cell stack to the load and it will also deliver a regulated output voltage [35]”.

line 307, the statement “The inductor current will increase linearly until reach a peak value of current and at this point, the voltage around the inductor will be equal to the input voltage source:” are revised in the same line of the revised manuscript as follows: “The inductor current will increase linearly until reaches a peak value of current and at this point, the voltage around the inductor will be equal to the input voltage source:”.

line 314, the statement “The previous equations can yield to the 15 which is the state-space equation that represents the boost converter dynamics [37].” are revised in the same line of the revised manuscript as follows: “The previous equations can yield the 15 which is the state-space equation that represents the boost converter dynamics [37]”.

Section 4, line 370, the statement “As previously enacted, the 16 refers to two terms that are related to each controller. In order to provide a differentiation in both designs, we define the 21 that establishes the control signal uc for the conventional STA.” are revised in the same line of the revised manuscript as follows: “As previously enacted, the 16 refers to two terms that are related to each controller. In order to provide differentiation in both designs, we define the 21 that establishes the control signal uc for the conventional STA.

line 383, the statement “The PI controller time domain can define as follows [43]:.” are revised in the same line of the revised manuscript as follows: “The PI controller time domain can be defined as follows [43]:”.

line 387, the statement “On the other hand, STA is robust and stable against perturbation and can overcome the chattering effects.” are revised in the same line of the revised manuscript as follows: “On the other hand, STA is robust and stable against perturbation and can overcome the chattering effect.”

line 401, the statement “By using equation (19), (20), (22) and (23),  differentiating equation (31) with respect to time yields to:” are revised in the same line of the revised manuscript as follows: “By using equation (19), (20), (22) and (23), differentiating equation (31) with respect to time yields the following:

line 403, the statement “The stability is achieved when both k1,2 are positive and major than 0. Consequently, according to the Lyapunov theory, the PEMFC power system is stable.” are revised in the same line of the revised manuscript as follows: “The stability is achieved when both k1,2 are positive and major than 0. Consequently, according to Lyapunov theory, the PEMFC power system is stable.”

Section 5, line 461, the statement “After one iteration, the black widows retained in the cannibalism stage and the black widows obtained in the mutation stage that are used as the initial population of the next iteration.” are revised in the same line of the revised manuscript as follows: “After one iteration, the black widows retained in the cannibalism stage and the black widows obtained in the mutation stage are used as the initial population of the next iteration.”.
